# Hierarchical Uncertainty Exploration via Feedforward Posterior Trees

**Elias Nehme**
Electrical and Computer Engineering
Technion–Israel Institute of Technology
`seliasne@campus.technion.ac.il`

**Rotem Mulayoff**
CISPA Helmholtz Center
for Information Security
`rotem.mulayof@gmail.com`

**Tomer Michaeli**
Electrical and Computer Engineering
Technion–Israel Institute of Technology
`tomer.m@ee.technion.ac.il`

## Abstract

When solving ill-posed inverse problems, one often desires to explore the space of potential solutions rather than be presented with a single plausible reconstruction. Valuable insights into these feasible solutions and their associated probabilities are embedded in the posterior distribution. However, when confronted with data of high dimensionality (such as images), visualizing this distribution becomes a formidable challenge, necessitating the application of effective summarization techniques before user examination. In this work, we introduce a new approach for visualizing posteriors across multiple levels of granularity using *tree*-valued predictions. Our method predicts a tree-valued hierarchical summarization of the posterior distribution for any input measurement, in a single forward pass of a neural network. We showcase the efficacy of our approach across diverse datasets and image restoration challenges, highlighting its prowess in uncertainty quantification and visualization. Our findings reveal that our method performs comparably to a baseline that hierarchically clusters samples from a diffusion-based posterior sampler, yet achieves this with orders of magnitude greater speed. Code and examples are available at our webpage.

## 1 Introduction

Communicating prediction uncertainty is a crucial element in advancing the reliability of machine learning models. This is particularly important in the realm of imaging inverse problems, in which a given input can typically be associated with a multitude of plausible solutions. In such cases, it is advantageous to equip users with efficient tools for exploring and visualizing the set of admissible solutions. Such tools may be of high value especially in safety-critical domains, such as scientific and medical image analysis [1–4], where inaccuracies in predictions could impact human lives.

Information about the plausible solutions and their respective likelihoods is encapsulated within the posterior distribution. However, high-dimensional posteriors are challenging to visualize. One popular approach for communicating uncertainty is to generate samples from the posterior [5–14]. Yet, in complex domains with high uncertainty levels, users may need to sift through hundreds of posterior samples per input to be able to confidently validate or refute suspicions about the unobserved ground-truth image [15]. Several methods have been proposed for generating a concise set of samples that highlight posterior diversity [15, 16]. However, choosing this set to be small is often insufficient for summarizing the posterior, while choosing it to be large hinders the user's ability to properly

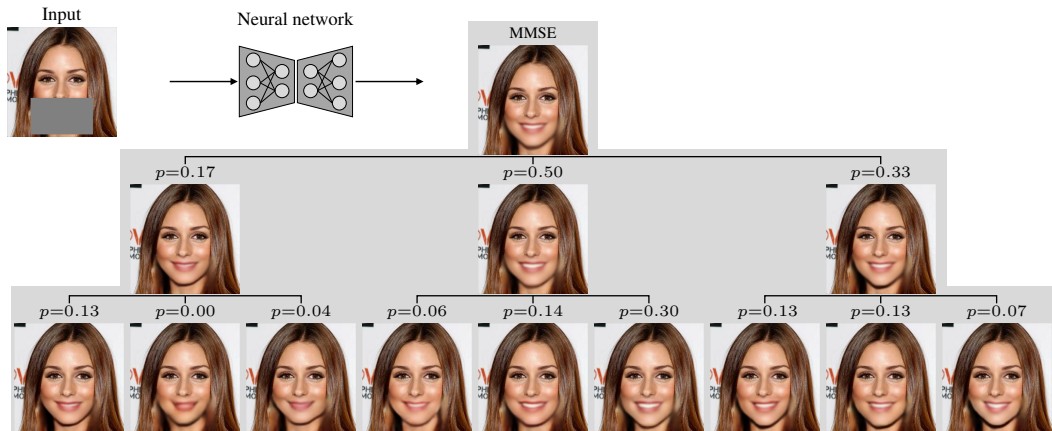

Figure 1: **Hierarchical decomposition of the minimum-MSE predictor into prototypes in the task of mouth inpainting**. The predicted tree explores the different options of bigger/smaller lips, mouth opening/closing, round/square jawline, etc.

inspect all solutions in the set. Other methods proposed to visualize uncertainty by allowing the user to navigate along the principal components (PCs) of the posterior [17–20]. While more interactive in nature, the projection onto a low dimensional principal subspace commonly accounts for only a small fraction of the reconstruction error.

In this work, we propose to model posterior uncertainty through *hierarchical* clustering, by using tree-structured outputs. In stark contrast to all existing methods, our approach supports efficient user interaction, allowing examination of only a small number of hypotheses en route to accepting or refuting a hypothesis about the unknown image. Specifically, our model receives a degraded measurement and outputs a hierarchy of predictions that cluster the solution set at multiple levels of granularity, each accompanied by their probability. This allows us to visualize the likelihood of different posterior modes, facilitating an informed visual exploration of posterior uncertainty (Fig. 1). Our method, which we coin *posterior trees*, is a generic technique for uncertainty visualization that is seamlessly transferable across tasks and datasets. Compared to the popular practice of training point estimators with mean square error (MSE) minimization, our method is an ideal drop-in replacement. This is because it adds a minor additional computational burden, while simultaneously accompanying the point estimate with its decomposition to the constituting clusters, revealing the underlying modes of variation and disclosing posterior uncertainty.

We showcase our method on multiple inverse problems in imaging, demonstrating the practical benefit of tree-valued predictions. We also quantitatively compare our learned posterior trees to a two-step baseline that generates samples using a diffusion-based posterior sampler and then applies hierarchical K-means clustering. Our approach achieves a comparable performance across tasks and datasets while enjoying a significantly faster runtime.

## 2   Related Work

The common avenue for probing data uncertainty (also known as aleatoric uncertainty [6–8, 21–23]) involves posterior sampling using conditional generative models like [5–14, 24–27], with the dominance of score-based/diffusion models [28–31] in recent years. While posterior sampling theoretically offers a sizable solution set per input, allowing useful uncertainty estimates through summarization methods like PCA [18] or $K$-means, its implementation proves impractically slow for contemporary state-of-the-art models. Despite attempts to enhance speed [32–35], the strategy remains plagued by extended run times.

In an effort to expedite inference, some studies proposed approximating posteriors with simpler distributions considering pixel correlations, such as correlated Gaussians [36–39]. More recently, two approaches emerged, directly outputting the top principal components (PCs) of the posterior distribution without imposing distributional assumptions [17, 19]. Additionally, subsequent work by

Yair et al. [20] visualized the *projected* posterior distribution onto the subspace spanned by the top PCs, unveiling its multi-modal nature at heightened uncertainty levels.

*Multiple choice learning* (MCL) [40–50] was proposed as a framework that can provide a multi-modal set of predictions when confronted with ambiguous inputs. The idea was originally proposed by Guzman-Rivera et al. [40] using structured support vector machines, and was later introduced to deep models in [42, 43]. The main working principle of MCL is to train an architecture with multiple output heads using the *oracle* loss, such that for a given input, only the head that is closest to the desired output gets updated. This winner-takes-all strategy encourages each head to specialize in a different mode of the output space, effectively predicting the centers of a Voronoi tessellation of all possible outputs [45]. Later works such as [48, 50] proposed to complement each prediction in the set with its likelihood. Nonetheless, existing multi-hypothesis methods are mainly tailored to classification settings, and are still limited in their ability to organize the predicted set of hypotheses effectively. Categorizing the solution set *hierarchically*, as we propose here, can significantly accelerate uncertainty exploration. For example, it allows iterative user interaction, focusing user efforts on the examination of a small number of hypotheses (not necessarily the most likely ones) to confirm/refute a suspicion about the signal. Moreover, the hierarchical structure can regularize the number of prototypes devoted to high-density modes, highlighting rare cases.

## 3 Method

Our goal is to predict a clean signal $\boldsymbol{x} \in \mathcal{X} \subseteq \mathbb{R}^{d_x}$, given some degraded measurements $\boldsymbol{y} \in \mathcal{Y} \subseteq \mathbb{R}^{d_y}$. We assume that $\boldsymbol{x}$ and $\boldsymbol{y}$ are realizations of random vectors $\mathbf{x}$ and $\mathbf{y}$ with an unknown joint distribution $p_{\mathbf{x},\mathbf{y}}(\boldsymbol{x}, \boldsymbol{y})$, from which we have a training set of *i.i.d.* samples $\mathcal{D} = \{(\boldsymbol{x}_i, \boldsymbol{y}_i)\}_{i=1}^N$. The objective of our predicted trees is therefore to visualize the uncertainty in the posterior distribution $p_{\mathbf{x}|\mathbf{y}}(\boldsymbol{x}|\boldsymbol{y})$ via a hierarchical set of a few prototypes.

### 3.1 Multiple Output Prediction

Many image restoration methods output a single prediction $\hat{\boldsymbol{x}}$ for any given input $\boldsymbol{y}$. A common choice is to aim for the posterior mean $\hat{\boldsymbol{x}} = \mathbb{E}[\mathbf{x}|\mathbf{y} = \boldsymbol{y}]$, which is the predictor that minimizes the MSE. However, a single prediction does not convey to the user the uncertainty in the restoration, especially if the solution set is multi-modal. When confronted with such ambiguous tasks, it is more natural to predict a small set of prototypical restorations $\{\hat{\boldsymbol{x}}_1, \ldots, \hat{\boldsymbol{x}}_K\}$ that portray the different options. In MCL, this is achieved by minimizing the so-called oracle/winner-takes-all loss given by

$$\mathcal{L}_{\mathrm{MCL}} \left( \boldsymbol{x}, \{\hat{\boldsymbol{x}}_i\}_{i=1}^K \right) = \min_{i=1,\ldots,K} \ell(\boldsymbol{x}, \hat{\boldsymbol{x}}_i), \tag{1}$$

where $\ell(\cdot, \cdot)$ is a loss function that measures prediction distance. Note that $\mathcal{L}_{\mathrm{MCL}}$ is minimized as long as one restoration in the predicted set is close to $\boldsymbol{x}$. This encourages prediction diversity, such that each restoration specializes in a different mode of the posterior. Rupprecht et al. [45] proposed a probabilistic interpretation of MCL, showing that the optimal minimizers of Eq. (1) form a centroidal Voronoi tessellation (CVT) of the posterior, where $\{\hat{\boldsymbol{x}}_i^\star\}_{i=1}^K$ are given by the conditional means/centroids of the resulting $K$ Voronoi cells (see Appendix E). In a discrete setting where we are given samples from $p_{\mathbf{x}|\mathbf{y}}(\boldsymbol{x}|\boldsymbol{y})$, CVT with the Euclidean distance is equivalent to $K$-means clustering.

Our goal here is to generalize the concept of predicting a set of restorations. Given an input $\boldsymbol{y}$, instead of predicting an unordered set $\{\hat{\boldsymbol{x}}_1(\boldsymbol{y}), \ldots, \hat{\boldsymbol{x}}_K(\boldsymbol{y})\}$, we propose to output an input-adaptive tree $\mathcal{T}(\boldsymbol{y})$ representing a hierarchical clustering of the posterior $p_{\mathbf{x}|\mathbf{y}}(\boldsymbol{x}|\boldsymbol{y})$. Specifically, we want the tree nodes to correspond to a hierarchical CVT of the posterior (analogous to a hierarchical $K$-means process in the discrete setting), where the children of each node constitute a CVT for the cluster represented by the node. Such a tree can organize the different restorations in a manner that facilitates their navigation by a user and can also accompany each node with its relative cluster probability.

### 3.2 Posterior Trees

As discussed above, training a model to predict a tree of degree $K$ and depth $d = 1$ with the oracle loss, results in a CVT of the posterior $p_{\mathbf{x}|\mathbf{y}}(\boldsymbol{x}|\boldsymbol{y})$ for every $\boldsymbol{y}$. Let $\{\mathcal{A}_k(\boldsymbol{y})\}_{k=1}^K$ denote the resulting Voronoi cells at $d = 1$ for some input $\mathbf{y} = \boldsymbol{y}$, such that $\cup_{k=1}^K \overline{\mathcal{A}_k(\boldsymbol{y})} = \mathcal{X}$, with $\mathcal{X} \in \mathbb{R}^{d_x}$ being the

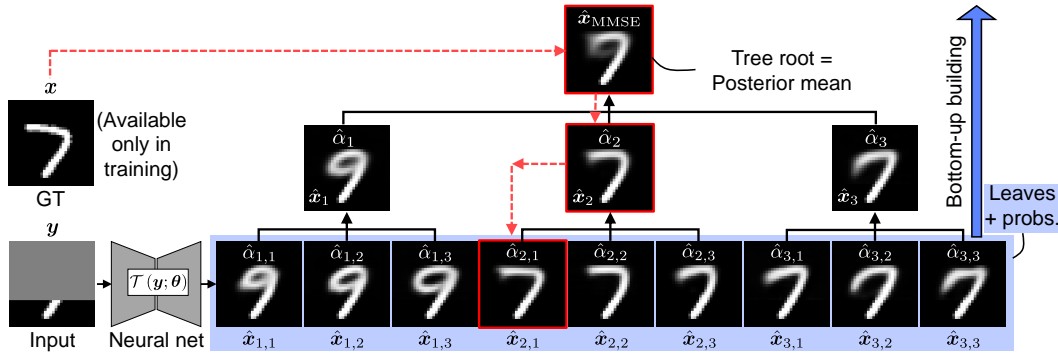

Figure 2: **Method overview**. Our model $\mathcal{T}(\boldsymbol{y};\boldsymbol{\theta})$ receives a degraded image $\boldsymbol{y}$ and predicts $\{\hat{\boldsymbol{x}}_{k_1,\ldots,k_d}\}_{k_1,\ldots,k_d=1}^{K}$, the bottom $K^d$ leaves, and their probabilities $\{\hat{\alpha}_{k_1,\ldots,k_d}\}_{k_1,\ldots,k_d=1}^{K}$ (faint blue box; illustrated here for $K=3$ and $d=2$). Next, the tree is iteratively constructed from the bottom up using weighted averaging, until we reach the root node which is the minimum MSE predictor $\hat{\boldsymbol{x}}_{\text{MMSE}}$. During training, starting from the root, the ground truth $\boldsymbol{x}$ is propagated through the tree until it reaches the leaves (dashed red lines). At tree level $d$, $x$ is compared to its immediate $K$ children nodes, and the MSE loss to the nearest child is added to the loss trajectory.

output space. Training with $\ell_2$ loss, *i.e.* $\ell(\hat{\boldsymbol{x}}, \boldsymbol{x}) = \|\hat{\boldsymbol{x}} - \boldsymbol{x}\|_2^2$, the optimal $K$ predicted leaves and probabilities are given by [45, 50]

$$
\begin{aligned}
\hat{\boldsymbol{x}}_k^\star(\boldsymbol{y}) &= \mathbb{E}\left[\mathbf{x}|\mathbf{y}=\boldsymbol{y}, \mathbf{x} \in \mathcal{A}_k(\boldsymbol{y})\right], \\
\hat{\alpha}_k^\star(\boldsymbol{y}) &= \mathbb{P}\left(\mathbf{x} \in \mathcal{A}_k(\boldsymbol{y})|\mathbf{y}=\boldsymbol{y}\right), \quad k = 1, \ldots, K.
\end{aligned}
\tag{2}
$$

We now want to extend the tessellation to the children of each node at the next level. Let $\{\mathcal{A}_{k,q}(\boldsymbol{y})\}_{q=1}^{K}$ denote the Voronoi cells at $d=2$, partitioning/tessellating the Voronoi cell $\mathcal{A}_k(\boldsymbol{y})$ from level 1 into $K$ sub cells such that $\cup_{q=1}^{K}\overline{\mathcal{A}_{k,q}(\boldsymbol{y})} = \mathcal{A}_k(\boldsymbol{y})$. We want the $K^2$ leaves and their associated probabilities outputted by our model to be given by

$$
\begin{aligned}
\hat{\boldsymbol{x}}_{k,q}^\star(\boldsymbol{y}) &= \mathbb{E}\left[\mathbf{x}|\mathbf{y}=\boldsymbol{y}, \mathbf{x} \in \mathcal{A}_{k,q}(\boldsymbol{y})\right], \\
\hat{\alpha}_{k,q}^\star(\boldsymbol{y}) &= \mathbb{P}\left(\mathbf{x} \in \mathcal{A}_{k,q}(\boldsymbol{y})|\mathbf{y}=\boldsymbol{y}\right), \quad k,q = 1, \ldots, K.
\end{aligned}
\tag{3}
$$

For simplicity of notation, in the following, we omit the dependence of the Voronoi cells on $\boldsymbol{y}$, and write $\mathcal{A}_k/\mathcal{A}_{k,q}$ instead. Equations (2) and (3) expose an interesting connection between tree levels. First, recall that since $\{\mathcal{A}_{k,q}\}_{k,q}$ is a tessellation, it satisfies $\cup_{q=1}^{K}\overline{\mathcal{A}_{k,q}} = \mathcal{A}_k$, and $\mathcal{A}_{k_1,q_1} \cap \mathcal{A}_{k_2,q_2} = \emptyset$ for $(k_1, q_1) \neq (k_2, q_2)$. Invoking the law of total expectation we can write

$$
\mathbb{E}\left[\mathbf{x}|\mathbf{y}=\boldsymbol{y}, \mathbf{x} \in \mathcal{A}_k\right] = \sum_{q=1}^{K} \mathbb{E}\left[\mathbf{x}|\mathbf{y}=\boldsymbol{y}, \mathbf{x} \in \mathcal{A}_k, \mathbf{x} \in \mathcal{A}_{k,q}\right] \mathbb{P}\left(\mathbf{x} \in \mathcal{A}_{k,q}|\mathbf{y}=\boldsymbol{y}, \mathbf{x} \in \mathcal{A}_k\right).
\tag{4}
$$

Equation (4) can be simplified by noting that $\mathcal{A}_k \cap \mathcal{A}_{k,q} = \mathcal{A}_{k,q}$, hence the expectation in the summand is given by $\mathbb{E}[\mathbf{x}|\mathbf{y}=\boldsymbol{y}, \mathbf{x} \in \mathcal{A}_{k,q}]$. Moreover, we further simplify the probability in Eq. (4) by writing

$$
\begin{aligned}
\mathbb{P}\left(\mathbf{x} \in \mathcal{A}_{k,q}|\mathbf{y}=\boldsymbol{y}, \mathbf{x} \in \mathcal{A}_k\right) &= \frac{\mathbb{P}\left(\mathbf{x} \in \mathcal{A}_k|\mathbf{y}=\boldsymbol{y}, \mathbf{x} \in \mathcal{A}_{k,q}\right) \mathbb{P}\left(\mathbf{x} \in \mathcal{A}_{k,q}|\mathbf{y}=\boldsymbol{y}\right)}{\mathbb{P}\left(\mathbf{x} \in \mathcal{A}_k|\mathbf{y}=\boldsymbol{y}\right)} \\
&= \frac{\mathbb{P}(\mathbf{x} \in \mathcal{A}_{k,q}|\mathbf{y}=\boldsymbol{y})}{\sum_{j=1}^{K} \mathbb{P}(\mathbf{x} \in \mathcal{A}_{k,j}|\mathbf{y}=\boldsymbol{y})},
\end{aligned}
\tag{5}
$$

where the first equality follows from Bayes' rule and the second equality is by the law of total probability and the fact that $\mathcal{A}_{k,q} \subseteq \mathcal{A}_k$, which implies that $\mathbb{P}(\mathbf{x} \in \mathcal{A}_k|\mathbf{y}=\boldsymbol{y}, \mathbf{x} \in \mathcal{A}_{k,q}) = 1$.

Substituting Eq. (5) and the simplified expectation in Eq. (4) with the notations of Eqs. (2) and (3), we obtain that

$$\hat{\boldsymbol{x}}_k^\star(\boldsymbol{y}) = \sum_{q=1}^{K} \hat{\boldsymbol{x}}_{k,q}^\star(\boldsymbol{y}) \frac{\hat{\alpha}_{k,q}^\star(\boldsymbol{y})}{\sum_{j=1}^{K} \hat{\alpha}_{k,j}^\star(\boldsymbol{y})}, \tag{6}$$

$$\hat{\alpha}_k^\star(\boldsymbol{y}) = \sum_{q=1}^{K} \hat{\alpha}_{k,q}^\star(\boldsymbol{y}), \quad k = 1, \ldots, K, \tag{7}$$

where Eq. (7) is by the law of total probability. Note that Eqs. (6) and (7) expose the redundancy in predicting both $\{\hat{\boldsymbol{x}}_k^\star(\boldsymbol{y}), \hat{\alpha}_k^\star(\boldsymbol{y})\}$ and $\{\hat{\boldsymbol{x}}_{k,q}^\star(\boldsymbol{y}), \hat{\alpha}_{k,q}^\star(\boldsymbol{y})\}$, as given the latter, we can compute the former in closed form. Furthermore, although our derivation assumed a tree of depth $d = 2$, the result trivially generalizes to a tree of depth $d$. Given the leaves and their probabilities at level $d$, $\{\hat{\boldsymbol{x}}_{k_1,\ldots,k_d}^\star(\boldsymbol{y}), \hat{\alpha}_{k_1,\ldots,k_d}^\star(\boldsymbol{y})\}_{k_1,\ldots,k_d=1}^{K}$, we can compose their parents at the upper $d-1$ levels, by employing Eqs. (6) and (7) iteratively from the bottom up.

Here, this is precisely the property we exploit to output an input-adaptive tree $\mathcal{T}(\boldsymbol{y})$ of degree $K$ and depth $d$, using a *single* model. Specifically, we train a model $\mathcal{T}(\boldsymbol{y}; \boldsymbol{\theta}) \triangleq (\hat{\boldsymbol{X}}_d(\boldsymbol{y}; \boldsymbol{\theta}), \hat{\boldsymbol{\alpha}}_d(\boldsymbol{y}; \boldsymbol{\theta}))$ that outputs $K^d$ prediction leaves $\hat{\boldsymbol{X}}_d(\boldsymbol{y}; \boldsymbol{\theta}) = \{\hat{\boldsymbol{x}}_n(\boldsymbol{y}; \boldsymbol{\theta})\}_{n=1}^{K^d}$ and their accompanying probabilities $\hat{\boldsymbol{\alpha}}_d(\boldsymbol{y}; \boldsymbol{\theta}) = \{\hat{\alpha}_n(\boldsymbol{y}; \boldsymbol{\theta})\}_{n=1}^{K^d}$ (see Fig. 2). Next, starting from the bottom leaves at depth $d$, the rest of the tree is composed level by level, recursively employing Eqs. (6) and (7) from the bottom up.

During training, the tree hierarchy is enforced through our loss function. Given some input $\boldsymbol{y}_i$, the model $\mathcal{T}(\boldsymbol{y}; \boldsymbol{\theta})$ outputs are used as explained earlier to construct the full tree, where the root of the tree approximates the posterior mean $\mathbb{E}[\mathbf{x}|\mathbf{y} = \boldsymbol{y}_i]$. Next, the label $\boldsymbol{x}_i$ is first compared to the tree root with an $\ell_2$ loss to ensure our decomposition recovers the MMSE estimator. At each successive level, the label $\boldsymbol{x}_i$ is compared to the children of the parent with minimal loss, and the closest child is added to the loss trajectory (Fig. 2). This process is repeated until we arrive at the bottom leaves.

Note that the described training scheme serves two purposes. First, it induces the desired tree structure by employing an amortized hierarchical version of the oracle loss in Eq. (1). Second, the loss employed on the root node enables predicting branch probability, mitigating the need for supervising the probabilities with explicit targets. See Appendix D for more details on our training algorithm and hierarchical oracle loss function.

### 3.3 Architecture

The approach presented in Section 3.2 is general, and can be used to augment any architecture outputting $K^d$ images and probabilities. Here, we choose to adapt the U-Net architecture [51] as a generic established choice for image-to-image regression tasks. The number of output images is $K^d$ to accommodate a tree of degree $K$ and depth $d$. In case all features are shared in the architecture, this amounts to changing the number of output filters at the last layer. However, in challenging tasks such as image inpainting, we found that sharing all parameters leads to reduced prediction diversity. In such cases, to trade off feature sharing and output diversity, we use group convolutions [52] in the decoder, such that each prediction has a disjoint set of channels learned separately from other outputs. As for the skip connections from the encoder, the concatenated features are interleaved equally per level such that each prediction has an equal share. In addition to the output images, our architecture also predicts $K^d$ probabilities (see Figs. 2 and A2). This is achieved by (global) average pooling all feature maps from the decoder, and feeding their concatenation to an additional lightweight *multi-layer perceptron* (MLP), with four linear layers and a softmax at the output.

### 3.4 Preventing Tree Collapse

Vanilla training with the oracle loss is known to suffer from "hypotheses collapse", where some of the predictions are implausible [40, 42, 43, 45, 46, 50, 53, 54]. This phenomenon occurs when some predictions are initialized worse than others in stochastic training, and therefore receive little to no gradient updates due to not being chosen in the oracle loss. This results in degenerate outputs that are not encouraged to produce meaningful results. Previous works proposed various regularizations to remedy this undesired effect. For example, Rupprecht et al. [45] relaxed the $\arg\min$ operator with a small constant $\varepsilon$, such that predictions with non-minimal loss get updated with $\varepsilon$-scaled gradients.

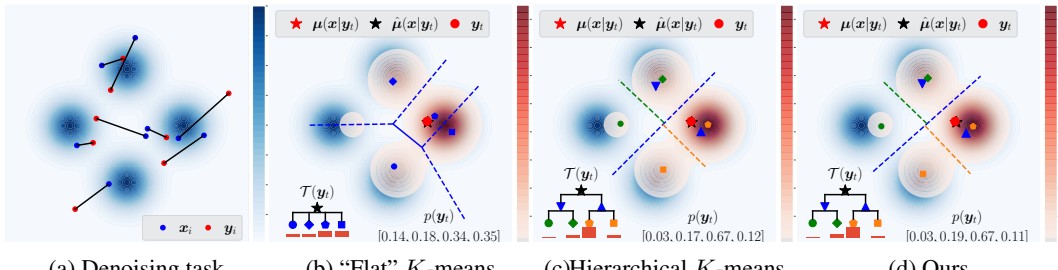

(a) Denoising task      (b) "Flat" $K$-means      (c)Hierarchical $K$-means      (d) Ours

Figure 3: **2D Gaussian mixture denoising**. (a) Underlying signal prior $p_\mathbf{x}(\boldsymbol{x})$ (blue heatmap), and training samples $(\boldsymbol{x}_i, \boldsymbol{y}_i) \sim p_{\mathbf{x},\mathbf{y}}(\boldsymbol{x}, \boldsymbol{y})$. (b) $K$-means with $K = 4$ applied to 10K samples $\boldsymbol{x}_i \sim p_{\mathbf{x}|\mathbf{y}}(\boldsymbol{x}|\boldsymbol{y}_t)$, for a given test point $\boldsymbol{y}_t$ (red circle). The resulting cluster centers (blue markers) partition the underlying posterior $p_{\mathbf{x}|\mathbf{y}}(\boldsymbol{x}|\boldsymbol{y}_t)$ (red heatmap), resulting in cluster probabilities $p(\boldsymbol{y}_t)$. (c) Hierarchical $K$-means applied twice with $K = 2$ on 10K samples $\boldsymbol{x}_i \sim p_{\mathbf{x}|\mathbf{y}}(\boldsymbol{x}|\boldsymbol{y}_t)$. At depth $d = 1$, the posterior is partitioned by the dashed blue line (blue triangles mark cluster centers). The resulting half spaces are subsequently halved by the dashed orange and green lines respectively. (d) Posterior trees (ours) with degree $K = 2$ and depth $d = 2$. Note that in all cases the estimated posterior mean $\hat{\boldsymbol{\mu}}(\boldsymbol{x}|\boldsymbol{y}_t)$ (black star) coincides with the analytical mean $\boldsymbol{\mu}(\boldsymbol{x}|\boldsymbol{y}_t)$ (red star), while in (c)-(d) the lowest density mode is better represented. $\mathcal{T}(\boldsymbol{y}_t)/p(\boldsymbol{y}_t)$ are drawn at the bottom of (b)-(d).

Moreover, Makansi et al. [54] proposed an evolving version of the oracle loss, where the top-$k$ predictions are updated in each step, with $k$ being annealed down to $k = 1$ during training, such that it starts with equal weights to all predictions and gradually shifts to updating only the best one towards convergence. Here, we employ a strategy that combines the simplicity of [45] with the adaptivity of [54]. Specifically, we scale the gradients of non-performing predictions with a constant $\varepsilon$ that is annealed during training according to

$$\varepsilon_t = \varepsilon_0 \exp\{-[t - t_0]_+/2\}, \tag{8}$$

where $t$ is the epoch number, and we fixed $\varepsilon_0 = 1$ and $t_0 = 5$ for all experiments. At the beginning of training, this has the benefit of bringing all predictions to a reasonable starting point, alleviating sensitivity to initialization and top prediction domination. As training progresses, this regularization is decayed, and only the relevant prediction is chosen for each sample, hence converging to the desired MCL behavior (see Appendix B).

### 3.5 Weighted Sampling

Albeit the regularization mentioned in Section 3.4, for tasks with highly imbalanced posteriors, training with Adam [55] leads to trees with a large disparity in leaf quality. This is because less likely leaves get chosen with lower frequency during training, resulting in transient gradients that highly affect the adaptive normalization in Adam. This hypothesis was also validated in matched settings by using SGD with an appropriate learning rate which resulted in significantly slower, yet improved convergence (see Appendix A.3). Therefore, to keep the speed advantage provided by Adam, in the task of image inpainting, we opted for a weighted sampler during training. The purpose of this sampler is to roughly balance out the overall number of occurrences at the bottom leaves during training. This is achieved by keeping track of an association matrix $\mathbf{A} \in \mathbb{R}^{K^d \times N}$, where $\mathrm{A}_{i,j}$ counts the number of times sample $\boldsymbol{x}_j$ was associated with leaf $c_i$ over some time window. This allows us to estimate the conditional probability $p_{\mathbf{c}|\mathbf{x}}(c_i|\boldsymbol{x}_j)$, and subsequently derive an optimized sample probability to achieve a roughly uniform marginal leaf probability across the entire training set. The loss of each sample is then adjusted to account for this intervention, avoiding tampering with the original posterior probabilities. See Appendix C for more details.

## 4 Experiments

Here we demonstrate our method in several settings. In all experiments except for the toy example, we used variants of the U-Net architecture [51], with a custom number of output images, adjusted to accommodate $K^d$ predictions according to the desired tree layout. Moreover, as explained in

Section 3.3, each model was also complemented with a small MLP for predicting leaf likelihood. Full details regarding the architectures, optimizer, and per-task settings are in Appendix A.

**Toy Example.**    As a warm-up, we demonstrate posterior trees on a 2D denoising task. Here, $\mathbf{x}$ is sampled from a mixture of four Gaussians (arranged in a rhombus-like layout), and $\mathbf{y}$ is a noisy version of $\mathbf{x}$. The prior distribution $p_{\mathbf{x}}(\boldsymbol{x})$ and exemplar samples from $p_{\mathbf{x},\mathbf{y}}(\boldsymbol{x},\boldsymbol{y})$ are presented in Fig. 3(a). For this simple case, the posterior distribution $p_{\mathbf{x}|\mathbf{y}}(\boldsymbol{x}|\boldsymbol{y})$ can be calculated analytically (see Appendix G for the derivation) and is also a mixture of Gaussians. Therefore, this task can serve as a sanity check enabling us to benchmark our results. To demonstrate our method on this toy example, we train a model $\mathcal{T}(\boldsymbol{y};\boldsymbol{\theta})$ outputting a tree of depth $d = 2$ and degree $K = 2$. The results are compared against an approximate "ground truth" in Fig. 3(c). Note that even though we have a closed-form expression for the posterior density, we still have to approximate the "ground truth" posterior tree by applying hierarchical K-means clustering to samples for every $\boldsymbol{y}$. As seen in Fig. 3(d), our method resulted in highly accurate posterior trees both in terms of cluster centers and cluster likelihoods. Moreover, this result was achieved while being presented only with a *single* posterior sample per $\boldsymbol{y}$ in training. In contrast, the approximate ground truth was computed with $10K$ samples. Moreover, it is important to note that besides trivial cases, our result could not have been achieved with a "flat" $K$-means clustering with $K = 4$ (Fig. 3(b)). This is because $K$-means tends to focus on high-density modes, whereas our hierarchical trees better convey rare cases of the posterior. See Appendix G for more details and examples.

**Handwritten Digits, Edges→Shoes, and Human Faces.**    Figure 2 demonstrates posterior trees on inpainting the top $70\%$ of handwritten digits from the MNIST dataset. As can be seen, at depth $d = 1$, the learned tree exposes the two likely modes averaged in the mean estimate $\hat{\boldsymbol{x}}$, being either a "7" or a "9". In addition, at deeper tree levels, the different modes are further refined to reveal intricate intra-digit variations. More examples are available in Appendix M. We also applied posterior trees to the edges-to-shoes dataset taken from pix2pix [56, 57]. Here, the task is to convert an image of black and white edges to an output RGB image of a shoe. As shown in Fig. 4(a), our tree is capable of representing diverse shoe colors, in an adaptive manner to the input contours.

Next, we tested posterior trees on face images from the CelebA-HQ dataset, using the split from CelebA [58]. Figure 4(b) demonstrates our method applied to image colorization, where the input is a grayscale image and the desired output is its RGB version. The resulting trees hierarchically refine the output predictions by background, skin tone, and hat color. We also tested our method on the task of image inpainting. Figures 1 and 4(c) show the resulting trees for mouth/eye inpainting respectively. For example, the predicted tree in Fig. 4(c) explores the different options of eye-opening/closing, eyebrow-raising/lowering, eyeglasses, etc. This demonstrates that even shallow trees can still depict diverse reconstruction characteristics, showcasing the benefit of outputting multiple predictions.

**Bioimage Translation.**    Transforming images from one domain to match the statistics of images from another is commonly referred to as the task of image-to-image translation [57]. In the realm of bioimaging, such transformations were utilized to predict fluorescence from bright-field images [1], "virtually stain" unstained tissue [3], and transfer the images of one fluorescent dye to appear as if they were imaged with another [59]. The common scenario in image-to-image translation is the task being highly ill-posed with a wide range of plausible transformations satisfying the desired statistics. For general image editing/style transfer, this is a desired property adding to the artistic excitement; However, in bioimaging, this is highly problematic as the result often informs a downstream task with high stakes, and therefore output uncertainty should be communicated.

Here, we applied our method to a dataset of migrating cells, imaged in a spinning-disk microscope, simultaneously with two different fluorescent dyes (one staining the nuclei and one staining actin filaments) [59]. The task was to predict the image of one fluorescent dye (nuclear stain) from another (actin stain). Figure 5 demonstrates the predicted tree for a $128 \times 128$ test patch. The tree conveys important information to the user exposing uncertain cells, and exploring optional cell shapes. These can for example affect downstream tasks such as cell counting and morphological cell analysis.

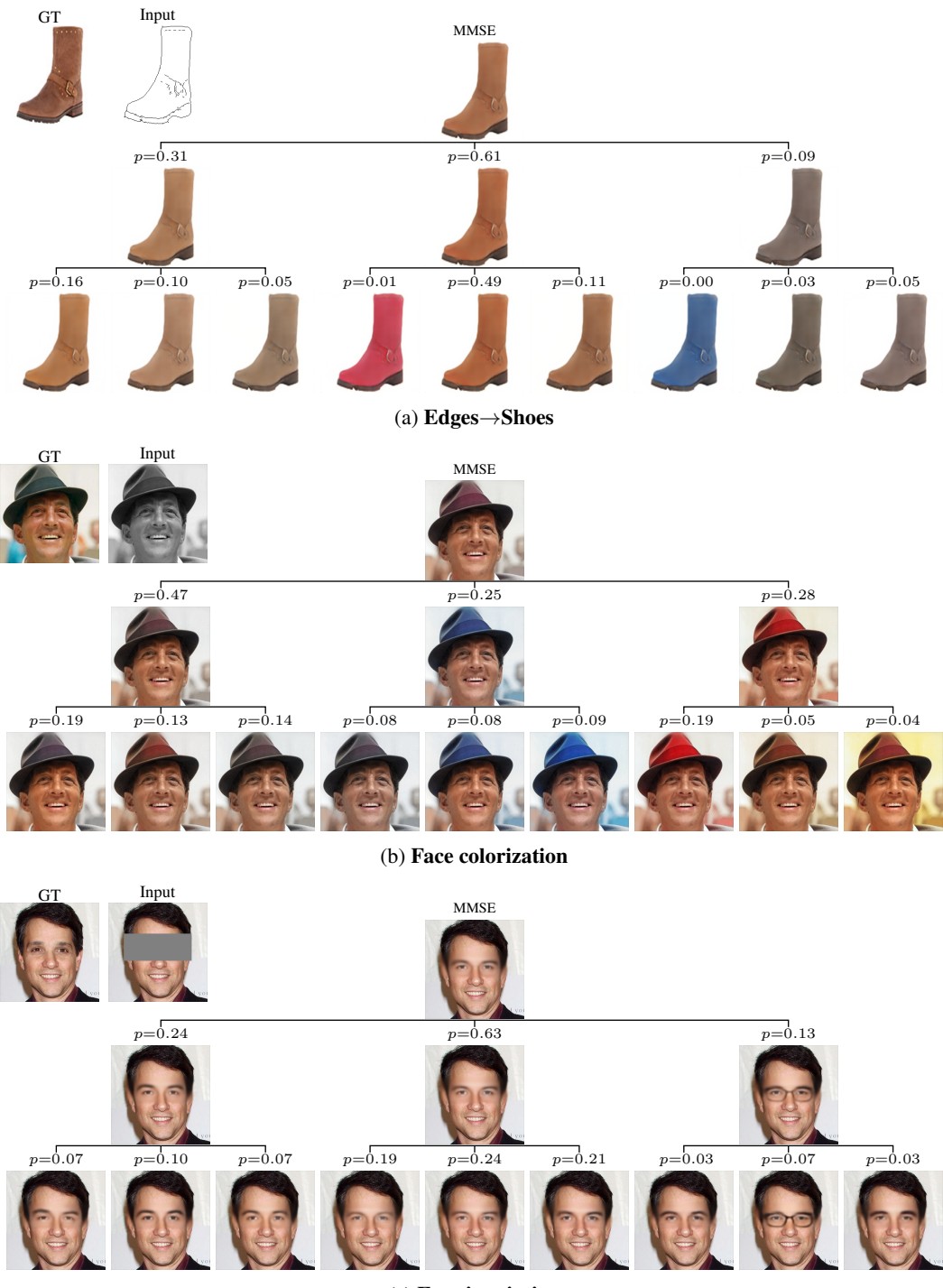

(a) **Edges→Shoes**

(b) **Face colorization**

(c) **Eyes inpainting**

Figure 4: **Diverse applications of posterior trees**. The predicted trees represent inherent task uncertainty: *e.g.,* (a) Refining the mean estimate by color, grouping similar colors, while still depicting unlikely ones (*e.g.,* the blue boot); (b) Presenting various plausible colorizations varying by hat color, skin tone, and background; and (c) Exploring the diverse options of eyebrows/eyeglasses.

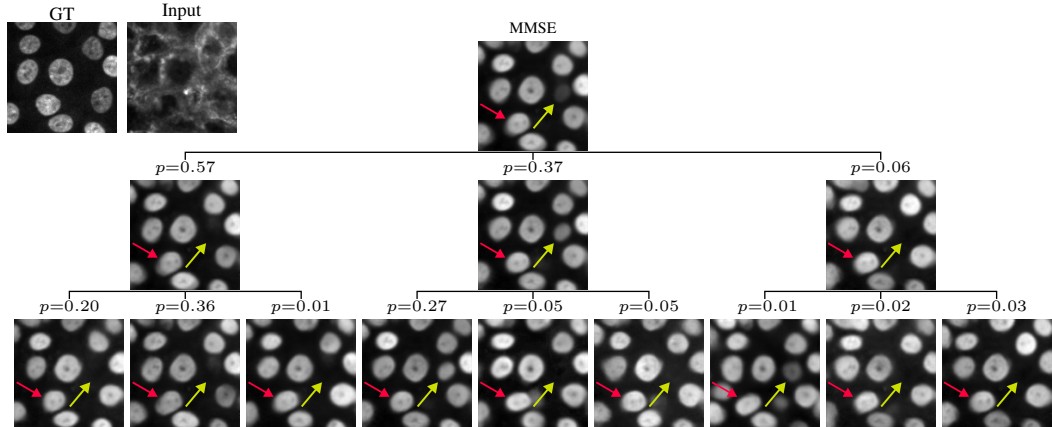

Figure 5: **Bioimage translation.** Here we explored posterior trees for the task of translating the image of a tissue from one fluorescent dye to another. The resulting trees expose important information regarding uncertain cells (yellow/red arrows), *e.g.,* ones that do not consistently appear in all branches, and additionally explore different plausible cellular morphology consistent with the input.

Table 1: Comparison to the proposed baseline on 100 test images from the FFHQ dataset. Hierarchical $K$-means was applied to 100 posterior samples per test image. Runtime is reported as both the speed of a forward pass (sec) and the memory usage (GB) required to infer a single test image with a batch of 1 on an A6000 GPU. The NLL at the root node ($d = 0$) is trivial and therefore omitted. Blue and Red indicate best and second best respectively.

| Task | Method | Optimal PSNR ($\uparrow$) | | | NLL ($\downarrow$) | | Speed | Memory |
|------|--------|-------|-------|-------|-------|-------|-------|--------|
| | | $d=0$ | $d=1$ | $d=2$ | $d=1$ | $d=2$ | (sec $\downarrow$) | (GB $\downarrow$) |
| Color. | DDNM | **24.6±3.8** | **25.5±3.6** | **26.1±3.6** | **0.9±0.4** | **2.0±0.6** | 340 | 18500 |
| | DDRM | **22.7±2.7** | 24.0±3.0 | 24.8±3.0 | **1.1±0.4** | **2.0±0.5** | **68** | **3700** |
| | Ours | **24.6±4.1** | **25.7±3.9** | **26.4±4.0** | **0.9±0.4** | **2.0±0.7** | **0.014** | **1.3** |
| Mouth inpaint | DDNM | 19.3±2.6 | **19.9±2.2** | 20.2±2.0 | 1.1±0.4 | 2.1±0.6 | 340 | 18500 |
| | RePaint | **19.8±2.7** | **20.5±2.3** | **20.7±2.3** | **1.0±0.3** | **1.9±0.4** | 15538 | 845450 |
| | MAT | 19.2±2.3 | 19.4±2.3 | 19.6±2.3 | 1.2±0.3 | 2.5±0.6 | **15** | **300** |
| | Ours | **20.1±2.4** | **20.5±2.3** | **20.4±2.2** | **0.9±0.4** | **2.0±0.8** | **0.014** | **1.3** |
| Eyes inpaint | DDNM | 19.3±2.7 | 19.6±2.5 | 19.5±3.7 | **1.1±0.7** | **2.1±0.6** | 340 | 18500 |
| | RePaint | **20.1±2.8** | **20.4±2.8** | **20.6±2.8** | **1.0±0.4** | **1.9±0.5** | 15538 | 845450 |
| | MAT | 19.0±3.0 | 19.2±3.0 | 19.3±3.0 | 1.2±0.3 | 2.4±0.6 | **15** | **300** |
| | Ours | **19.6±2.7** | **19.9±2.5** | **19.8±2.4** | 1.3±0.6 | 2.5±0.7 | **0.014** | **1.3** |

## 4.1 Quantitative Comparisons

**Baseline.** Building on the notion of clustering from Section 3.1, we propose a simple baseline to benchmark our results. As shown in Fig. 3(d), our predicted trees are constructed out of prototypes that yield a hierarchical clustering of the posterior. In discrete settings, this is equivalent to applying Hierarchical $K$-means on samples from $p_{\mathbf{x}|\mathbf{y}}(\boldsymbol{x}|\boldsymbol{y})$. Therefore, given a method to sample from the posterior (*e.g.,* using [7, 8, 10, 11]), a natural baseline in our case is a two-step procedure: (i) Generate $N_s$ samples from the posterior, and (ii) Apply hierarchical $K$-means $d$ times to generate a tree of degree $K$ and depth $d$. Note that due to computational reasons, the comparison is performed over a random subset of 100 test images from FFHQ, where for each image we generate $N_s = 100$ samples, and apply hierarchical $K$-means with $K = 3$ and $d = 2$. To ensure a fair comparison, at each clustering step, $K$-means was run 5 times, keeping only the clusters with the best objective.

**Metrics.** To compare our predicted trees to the proposed baseline, we opted for two different metrics. The first metric is the PSNR (equivalent to MSE) between the ground truth test image and the tree nodes along the optimal path starting from the root and ending at the leaves. Intuitively, an accurate posterior clustering implies our tree nodes should maximize the PSNR, representing the ground truth test image with increasingly higher accuracy (lower MSE) as a function of (optimal)

node depth. The second metric we adopt is the sample negative log-likelihood (NLL) (using the natural logarithm) of the ground truth test image under the predicted posterior partitioning. Here, an accurate tree is expected to maximize the sample log-likelihood. This metric serves the purpose of verifying the predicted probabilities, as in practice we do not have access to the ground truth posterior distribution, and estimating cluster probability based on posterior samples becomes worse as a function of depth. Table 1 compares posterior trees to the proposed baseline, implemented with various state-of-the-art samplers. The results suggest that our method yields comparable performance in both the PSNR along the optimal path and in sample log-likelihood while requiring a single forward pass ($\approx 14$ ms with a 1.3 GB memory footprint on a A6000 GPU). This is $10^3 - 10^7 \times$ faster than the competition without considering our wide advantage in memory footprint. In case we do consider a batched setting with a sizable test set, our advantage becomes even more pronounced, where our method enables the inference of $\approx 430$ test images in a single second. See Appendices I to L for more details and visual comparisons.

## 5   Discussion and Conclusion

We demonstrated the wide applicability of posterior trees across diverse tasks and datasets. However, our method is not free of limitations. First, the optimal hyper-parameters $K$ and $d$ are task-dependent, with no rule of thumb for determining them a-priori (see Appendix H). Second, in this work, we focused on balanced trees with a fixed degree $K$ which might be sub-optimal for certain posteriors. Devising a strategy for an input-adaptive tree layout is an exciting direction for future research. Third, our method is limited in the number of output leaves, as we amortize the entire tree inference to a single forward pass. For predicting significantly deeper trees, an iterative inference procedure conditioning the model on the node index (in an analogous fashion to the timestep in diffusion models) is required. Finally, our method is tailored towards visualizing uncertainty and not sampling realistic-looking images. Although with increased depth our prototypes become increasingly sharper, they are nonetheless still cluster centers that average multiple plausible solutions and hence are not expected to lie on the image data manifold for shallow trees. A possible solution is to apply posterior trees in the latent space of an autoencoder such as VQ-VAE [60, 61], however, this is beyond the scope of this paper.

To conclude, in this work, we proposed a technique to output a hierarchical quantization of the posterior in a single forward pass. We discussed key design choices underlying our approach, including bottom-up tree construction, a principled training scheme, and proper regularization techniques to prevent tree collapse. We further demonstrated the benefit of hierarchical clustering over flat trees and discussed the intuition behind it on a toy example. In our experiments, we applied our method to highly ill-posed inverse problems and showed that diverse (prototypical) reconstructions are possible with a simple training scheme exposing uncertainty. Additionally, we also proposed an appropriate baseline based on posterior sampling, and quantitatively compared our approach to several strong samplers. Our method demonstrated at least comparable results while being orders of magnitude faster. Finally, we applied our method to the challenging task of bioimage translation, demonstrating its practical relevance in a real-world application.

## Acknowledgments and Disclosure of Funding

This research was partially supported by the Israel Science Foundation (ISF) under Grant 2318/22 and by a gift from Elbit Systems. This work was done while Rotem Mulayoff was at the Technion. The authors also thank Hila Manor and Noa Cohen for their help with baseline comparisons.

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

# Appendices

## A    Experimental Details

### A.1    Architectures

As mentioned in Section 3.3, in our experiments we adopted the U-Net [51] architecture. Our architecture consisted of 4 downsampling/upsampling blocks. Downsampling by $2\times$ was performed using average pooling, and upsampling by $2\times$ was implemented with a nearest-neighbor interpolation. In both cases, the feature maps at the updated spatial resolution were processed by 2 convolution blocks consisting of 2D convolution, group normalization, and LeakyReLU activation with a negative slope of 0.2. The number of features at each spatial resolution was adapted to the predicted tree layout, *i.e.,* for predicting a tree of degree $K$ and depth $d$, the initial number of channels before downsampling was set to $c_{\text{init}}K^d$, where $K^d$ is the number of output leaves and $c_{\text{init}} \in \{4, 8\}$. At each successive encoder block, the number of channels was doubled, reaching $16c_{\text{init}}K^d$ at the bottleneck. For example, assuming a tree with degree $K = 3$ and depth $d = 2$, and setting $c_{\text{init}} = 4$, the number of channels per level in the encoder is given by $[36, 72, 144, 288, 576]$.

Similarly, in the decoder, the number of channels was halved at each upsampling step in a symmetric fashion. However, as mentioned in Section 3.3, in some tasks (*e.g.,* image inpainting), we noticed that fully sharing parameters between all leaves led to reduced diversity between predictions (see Fig. A1). On the other hand, learning a separate U-Net for each prediction leaf is computationally intensive, and highly inefficient as it is expected that the initial feature extraction stage for predicting the different leaves would be similar. Therefore, as a middle ground between these two extremes, we opted for an architecture that shares the encoder between the different leaves, while having disjoint decoders, each with a dedicated set of weights $\{\varphi_n\}_{n=1}^{K^d}$, learned separately from others. As for the skip connections from the encoder, the concatenated feature maps are interleaved equally per level such that each prediction has an equal share.

In addition to the output images, our architecture also predicts $K^d$ scalars, which are the probabilities of the different leaves. This is achieved by global average pooling of all feature maps from the decoder, and feeding their concatenation to an additional lightweight MLP. This MLP has four linear layers with dimensions $[d_f, 256, 64, K^d]$, where $d_f$ is the dimension of the concatenated pooled features from the decoder. Each linear layer is followed by a 1D batch normalization and SiLU non-linearity, and the output is passed through a $\mathrm{softmax}$ layer to produce a valid probability vector. Figure A2 summarizes our architecture.

### A.2    Per-task Details

In all tasks, we only learned the required residual from the input to produce the predictions. For MNIST experiments, the images were padded to $32 \times 32$ to enable proper downsampling in the encoder. For Edges→Shoes experiments, the images were kept the same as in the pix2pix [57] paper, with a resolution of $256 \times 256$. For CelebA-HQ experiments, the images were resized to $256 \times 256$. Finally, for the bioimage translation dataset, we trained on $128 \times 128$ patches cropped from the full $1024 \times 1024$ images, as cell information tends to be local.

### A.3    Optimization

We used the Adam optimizer [55] with $\beta_1 = 0.9, \beta_2 = 0.999$ for all experiments. For the U-Net predicting the output leaves, we used an initial step size of 0.001. For the MLP predicting leaf probability, however, we found it beneficial to use a smaller initial step size of 0.0002. This is because at the initial training phase, the predicted leaves are still not converged; hence, the MLP can easily "classify" which leaf is the most likely. This collapses the predicted probabilities to a sparse vector, leading to leaves with zero probability. Hence, to avoid this instability, we used a lower step size for the probabilities such that the leaves are first allowed to converge, leading to the desired learning dynamics. For both components, the step size was reduced by a factor of 10 if the validation loss stagnated for more than 10 epochs, and the minimum step size was set to $5 \cdot 10^{-6}$. We used a batch size of 32 for 70 epochs for all tasks. This resulted in training times of $\approx$40 mins, 10 hrs, 5 hrs, and 7 hrs for MNIST, Edges→Shoes, CelebA-HQ, and the bioimage datasets respectively.

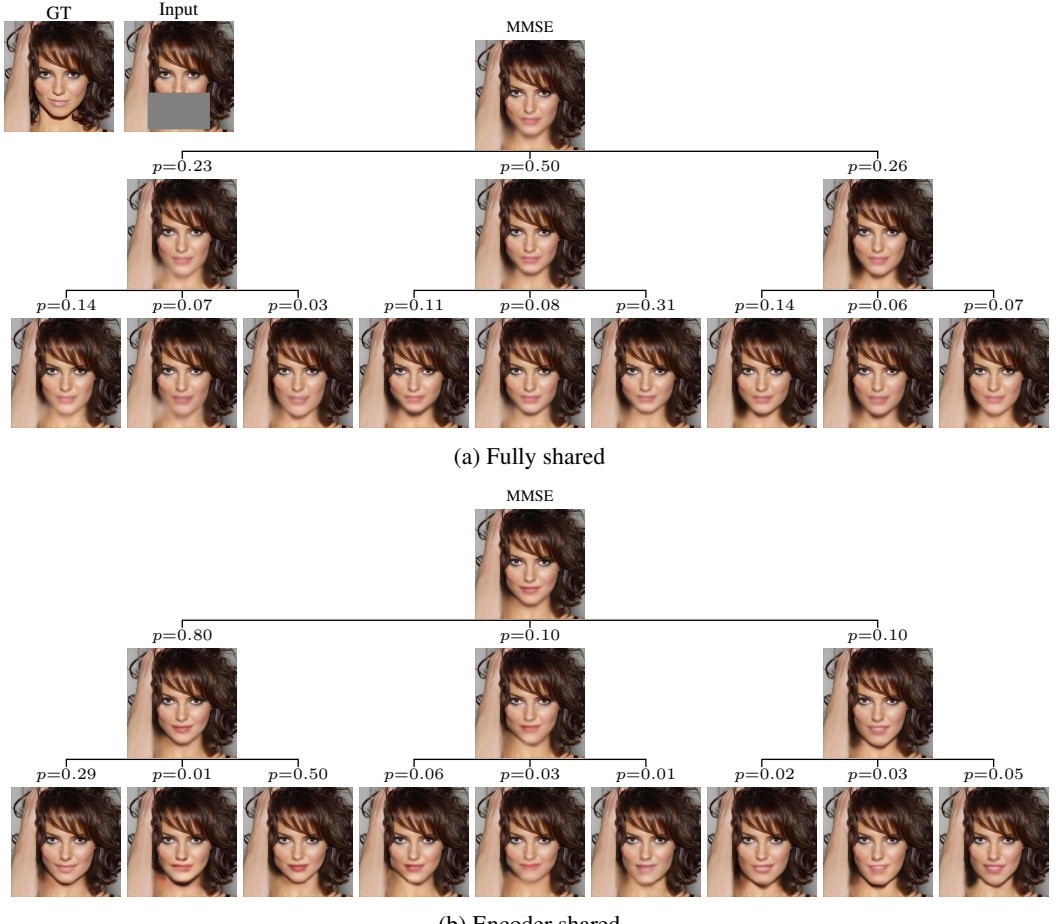

(a) Fully shared

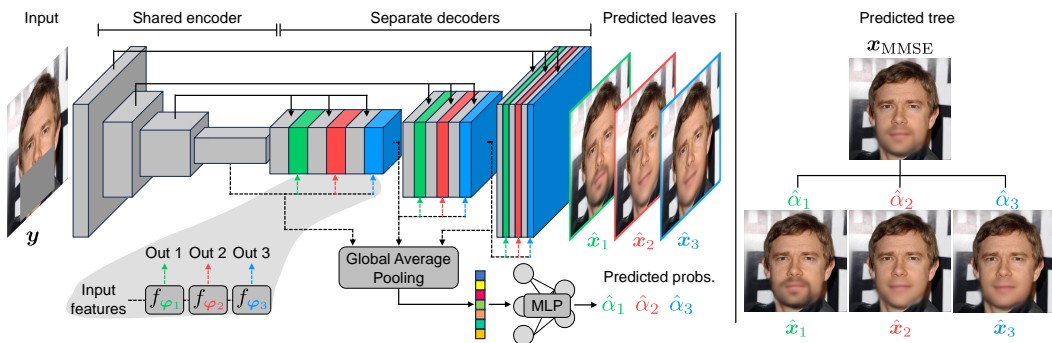

(b) Encoder shared

Figure A1: **Leaf weight sharing strategy.** (a) Fully shared architecture, with all leaves predicted jointly. (b) Leaves only share encoder (see Fig. A2).

Figure A2: **Model architecture**. Our model receives a degraded image $y$ and predicts the bottom $K^d$ leaves and their probabilities $\{\hat{\boldsymbol{x}}_{k_1,\ldots,k_d}, \hat{\alpha}_{k_1,\ldots,k_d}\}_{k_1,\ldots,k_d=1}^K$ (illustrated here for $K = 3$ and $d = 1$). The encoder is shared between all leaves, however, in the decoding stage, each leaf has a separate decoder (*e.g.,* with parameters $\varphi_1, \varphi_2, \varphi_3$) to enable diverse outputs. To predict leaf probability, all features from the decoder are passed through a global average pooling layer, and the resulting concatenated vector is passed to a lightweight MLP with a softmax layer at the output. Afterward, as explained earlier, the output tree is constructed iteratively from the bottom up (right).

# B Role of $\varepsilon_t$ From Eq. (8)

As explained in Section 3.4, the convergence of MCL is strongly affected by initialization. This is because leaves that are better initialized are more likely to be chosen in the oracle loss, and hence will dominate the remaining leaves in training. This results in output leaves that practically do not train, and therefore give meaningless results at test time. To remedy this, we scale the gradients of non-performing predictions with a constant $\varepsilon$ that is annealed during training according to Eq. (8). For the first $t_0 = 5$, our training degenerates to standard MSE minimization, bringing all leaves to a reasonable starting point near the posterior mean (Fig. A3(c)). This results in all prediction leaves having roughly the same initialization, with a similar number of associated training samples. Afterward, $\varepsilon$ is gradually decayed, and the leaves start to specialize in their respective posterior mode, converging to the desired MCL behavior (Fig. A3(d)). In this research, we identified this strategy as a simple yet effective method of regularization. However, it is important to highlight that a parallel study by Perera et al. [62] presents a potentially improved solution that combines soft assignments with deterministic annealing. This new approach could be easily integrated with our method, and it would be intriguing to explore this in future work.

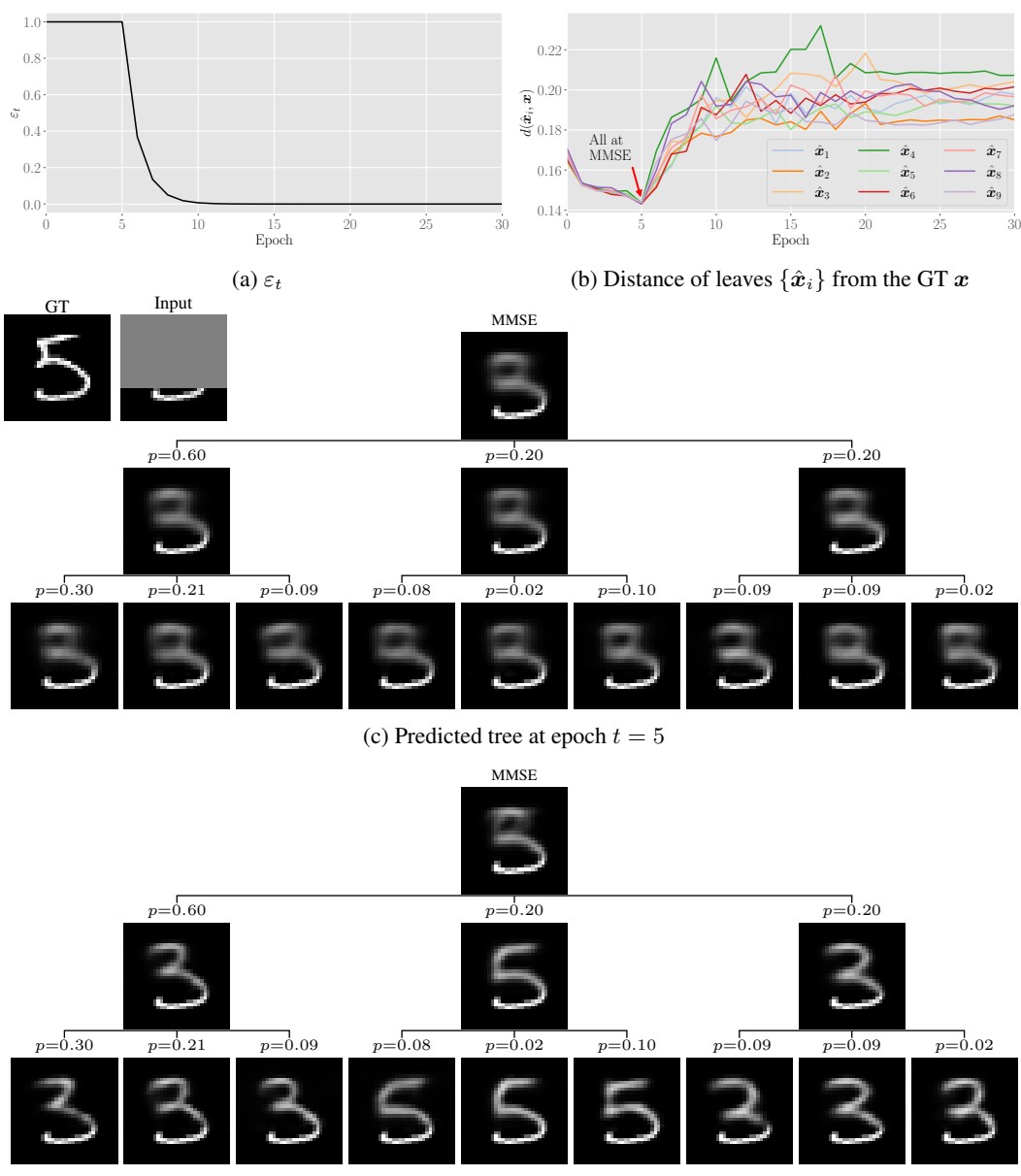

(a) $\varepsilon_t$

(b) Distance of leaves $\{\hat{x}_i\}$ from the GT $x$

(c) Predicted tree at epoch $t = 5$

(d) Predicted tree at convergence (epoch $t = 22$)

Figure A3: **Role of** $\varepsilon_t$. (a) $\varepsilon_t$. (b) Distances of leaf predictions $\{\hat{x}_i\}$ from the GT $x$ throughout epochs. For the first $t_0 = 5$ epochs all predictions are brought to the vicinity of the MMSE estimator. Afterward, $\varepsilon_t$ is decayed and each of the leaves is free to specialize in a subset of the posterior, leading to higher distances on average for the entire test set. (c) Predicted tree at epoch $t = t_0 = 5$ where all leaves are near the MMSE estimator. (d) Predicted tree at convergence (epoch $t = 22$) where each leaf specializes in a different posterior mode (*e.g.,* a "3" or a "5").

## C Weighted Sampler

For imbalanced posteriors where the dominant mode is significantly more likely than the weakest mode (*e.g.,* $10\times$), we found that an additional regularization complementary to the one provided by $\varepsilon_t$ is needed. This is because even if we eliminate the dependence of training on leaf initialization, still, leaves associated with (much) less likely posterior modes will be chosen with a lower frequency during training, resulting in transient gradients that highly affect the adaptive normalization in Adam. To test this hypothesis, we trained two models on the task of image inpainting: (i) one with Adam using an initial learning rate of 0.001, and (ii) one with SGD using a momentum of 0.9 and an initial learning rate of 0.1. Figure A4 demonstrates the results of this experiment. As evident in the resulting trees, Adam leads to highly implausible predictions in leaves with low likelihoods (*e.g.,* $p = 0.0$ in Fig. A4(c)). SGD, On the other hand, does not lead to nonsensical predictions; however, it requires significantly longer training time and is more challenging to converge (Fig. A4(d))

Therefore, in our method, we proposed a novel non-stochastic weighted sampling scheme as a simple fix (Fig. A4(e)). The proposed weighted sampler enables us to still enjoy the convergence speed of Adam while tackling the aforementioned optimization deficiency of MCL training for posteriors that are far from uniform. The goal of our sampler is to ensure that on average the number of occurrences at each output leaf during training is roughly the same (Fig. A4(b)). This is done by undersampling training samples associated with leaves that represent probable posterior modes while oversampling training samples associated with leaves that represent rare posterior modes.

Formally, let $c_i$ denote the $i$th prediction leaf ($i = 1, \ldots, K^d$), and $s_j = (x_j, y_j)$ denote the $j$th (paired) training sample. We assume we are given a training set $\mathcal{D} = \{s_j\}_{j=1}^N$ of $N$ *i.i.d.* samples, such that $p_s(s_j) = \frac{1}{N}$. The goal of the sampler is to manipulate $p_s(s_j)$ via oversampling/undersampling, such that the new sample probability in training $q_s(s_j)$, leads to a uniform marginal leaf distribution, *i.e.,* $q_c(c_i) = \frac{1}{K^d}$. To estimate the marginal leaf probability during training, we keep track of an association matrix $A \in \mathbb{R}^{K^d \times N}$, where $A_{i,j}$ counts the number of times sample $s_j$ was associated with leaf $c_i$. Specifically, after each training batch $b_\tau$ of size $B = |b_\tau|$, the association matrix is updated with momentum $\mu = 1 - 2^{-\frac{B}{N}}$, such that

$$A_0^{\text{EMA}} = 0, \qquad A_\tau^{\text{EMA}} = \mu A_{\tau-1}^{\text{EMA}} + (1 - \mu) A_\tau, \tag{A1}$$

where $A_\tau$ is a binary matrix that encodes the sample-leaf association for the batch $b_\tau$. This smoothing is necessary to accumulate statistics across epochs, making sample probability change gracefully while avoiding large changes to the probability of uncertain samples that switch leaf association between batches. Next, given the updated association matrix $A_\tau^{\text{EMA}}$, we can normalize it to obtain an estimate of the current joint distribution $p_{c,s}(c_i, s_j)$ as

$$p_{c,s}(c_i, s_j) = \frac{A_{i,j}^{\text{EMA}}}{\sum_{p=1}^{K^d} \sum_{\ell=1}^N A_{p,\ell}^{\text{EMA}}}. \tag{A2}$$

Manipulating the sample probability $p_s(s_j)$ will affect this statistic. Here we can assume that the conditional distribution over the leaves $\{c_i\}_{i=1}^{K^d}$ given a sample $s_j$ stays the same under the change of distribution for the samples. This insight can then be leveraged as follows. First, we obtain the current conditional leaf probability via marginalization

$$p_{c|s}(c_i|s_j) = \frac{p_{c,s}(c_i, s_j)}{\sum_{i=1}^{K^d} p_{c,s}(c_i, s_j)}. \tag{A3}$$

Let $P \in \mathbb{R}^{K^d \times N}$ denote the matrix where $P_{i,j} = p_{c|s}(c_i|s_j)$. Recall that our goal is to dictate a new sample probability $q_s(s_j)$ such that the induced marginal leaf probability $q_c(c_i)$ is uniform. Let $Q^{c|s} \in \mathbb{R}^{K^d \times N}$ denote the conditional distribution matrix under probability $q$ such that $Q_{i,j}^{c|s} = q_{c|s}(c_i|s_j)$. In matrix form, we are looking for a probability vector over the samples $q^s \in \mathbb{R}^N$, that satisfies $Q^{c|s} q^s = q^c = \frac{1}{K^d} \mathbf{1}_{K^d \times 1}$. Since the conditional probability of the leaves is assumed to be not affected by the change from $p_s$ to $q_s$, we have that $Q^{c|s} = P$. Therefore, in simpler notations, we are searching for a probability vector $q \in \mathbb{R}^N$ for which

$$Pq = \frac{1}{K^d} \mathbf{1}_{K^d \times 1}. \tag{A4}$$

Naturally, since $K^d \ll N$, there are infinitely many different options $\boldsymbol{q}$ that satisfy this constraint. Note that some of them can correspond to an imbalanced distribution, where some samples associated with a certain leaf have high probability, while other samples that are associated with the same leaf have low probability. Therefore, among all the different options that satisfy Eq. (A4), we want the one that has the flattest probability. The flattest distribution minimizes the $\ell_2$ loss, namely $\|\boldsymbol{q}\|_2^2$. However, minimizing this objective under the constraint in Eq. (A4) can be problematic for ill-conditioned cases. Nevertheless, we can achieve similar results by solving the following Tikhonov-regularized relaxed problem:

$$\min_{\boldsymbol{q} \in \mathbb{R}^N} \quad \frac{1}{2}\|\boldsymbol{P}\boldsymbol{q} - \frac{1}{K^d}\mathbf{1}\|_2^2 + \frac{\lambda}{2}\|\boldsymbol{q}\|_2^2$$
$$\text{s.t.} \quad q_j \geq 0, \quad j = 1, \dots, N \tag{A5}$$
$$\boldsymbol{q}^\top \mathbf{1}_{N \times 1} = 1$$

One can obtain an exact solution $\boldsymbol{q}^\star$ to Eq. (A5) using standard methods for convex optimization, *e.g.* algorithms in CVXPY [63]. However, in typical settings, this results in longer training times. Therefore, we opted to use an approximation. Specifically, if we drop the positivity constraint $q_j \geq 0, \forall j$, and solve Eq. (A5) accounting only for the constraint $\boldsymbol{q}^\top \mathbf{1}_{N \times 1} = 1$, we obtain the following closed-form solution

$$\tilde{\boldsymbol{q}} = \left( \boldsymbol{I}_{N \times N} - \boldsymbol{P}^\top \left( \boldsymbol{P}\boldsymbol{P}^\top + \lambda \boldsymbol{I}_{K^d \times K^d} \right)^{-1} \boldsymbol{P} \right) \mathbf{1}_{N \times 1}$$
$$\boldsymbol{q}^\diamond = \frac{1}{\tilde{\boldsymbol{q}}^\top \mathbf{1}_{N \times 1}} \tilde{\boldsymbol{q}}. \tag{A6}$$

In practice, in most cases, the positivity constraints are inactive, and therefore $\boldsymbol{q}^\diamond = \boldsymbol{q}^\star$. In the rare cases where Eq. (A6) resulted in negative values, we clipped entries below zero in $\boldsymbol{q}^\diamond$ and renormalized the result.

Given the desired sampling distribution $q_{\mathbf{s}}(\boldsymbol{s}_j)$ we pick for the next batch $\boldsymbol{b}_{\tau+1}$ the $B$ distinct samples that their selection will minimize the maximal distance between the future statistics at time $\tau + 1$ and $q_{\mathbf{s}}$. Specifically, at each training step, the next batch of samples is chosen such that the maximal difference between the next iteration estimated sample probability $p_{\mathbf{s}}(\boldsymbol{s}_j) = \sum_{i=1}^{K^d} p_{\mathbf{c},\mathbf{s}}(c_i, \boldsymbol{s}_j)$ and the desired sample probability $q_{\mathbf{s}}(\boldsymbol{s}_j)$, *i.e.*, $\|p_{\mathbf{s}} - q_{\mathbf{s}}\|_\infty$, is minimized. In practice, fetching the samples for future batches is done in advance to speed up the training time. Therefore, to prevent the training from waiting for the successor samples, we use a buffer of $m$ batches ahead and adjust the implementation accordingly. During training, we activated our weighted sampler one epoch after $\varepsilon_t$ started decaying to 0.

Finally, note that switching from $p_{\mathbf{s}}$ to $q_{\mathbf{s}}$ during training effectively changes the dataset by repetition/omission, and if not accounted for will distort the learned posterior probabilities. Therefore, to undo this undesired effect while still maintaining an increased number of optimization steps for weak posterior modes, we scaled the loss of sample $\boldsymbol{s}_j$ by the factor $\gamma_j = \frac{1}{N q_{\mathbf{s}}(\boldsymbol{s}_j)}$. Hence, we effectively only changed the number of optimization steps taken per output leaf, enabling all leaves to continuously receive gradients and train equally as well with Adam.

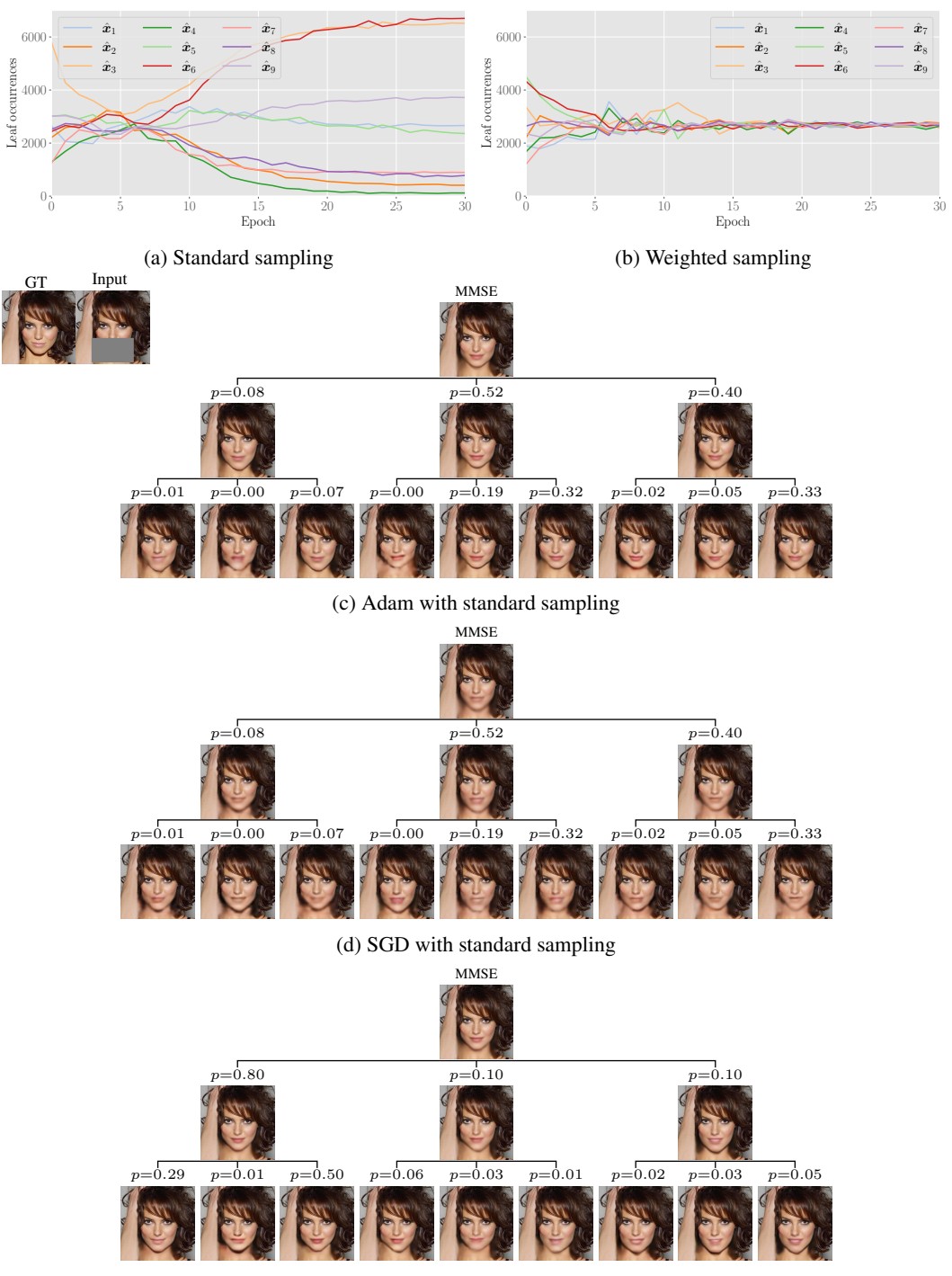

Figure A4: **Weighted sampling effect.** Optimization with Adam requires weighted sampling to train properly (see text in Appendix C).

# D Tree Loss Function and Training Scheme

Our tree training algorithm is summarized in Algorithm 1. Throughout this work, we used the MSE loss $\ell(\boldsymbol{x}, \hat{\boldsymbol{x}}) = \|\boldsymbol{x} - \hat{\boldsymbol{x}}\|_2^2$ to both determine the closest tree node and as our optimization loss. However, in general, a distinction should be made between the measure $\ell(\cdot, \cdot)$ used for the clustering (i.e. determining the associations of samples to clusters) and the loss used within each cluster to determine the cluster representative. In theory, our approach can work without modification with any association measure (e.g. LPIPS [64], some domain-specific classifier [65], etc.). However, changing the within-cluster loss requires some modifications. Specifically, the use of the MSE loss is what provides us with the hierarchical decomposition of the posterior mean (see Eqs. (2) to (7)). In particular, when using the MSE/$L_2$ loss, each cluster representative becomes the posterior cluster mean, and a weighted combination of those representatives gives the overall posterior mean (which is the tree root). This allows us to have the network output only the leaves of the tree, which implicitly defines the entire tree (as the nodes of each level are obtained as linear combinations of the nodes of its children).

One natural extension to investigate using our approach is to take the association measure to be $\ell(\hat{\boldsymbol{x}}, \boldsymbol{x}) = \|f(\hat{\boldsymbol{x}}) - f(\boldsymbol{x})\|_2^2$, where $f(\cdot)$ is some relevant domain-specific feature extractor. Specifically, we experimented with the deep features of the AnyCostGAN attribute predictor [65], and employed it in the task of eyes inpainting using the CelebAHQ dataset (Fig. A5). This preliminary result indicates that our method could potentially be used with other association metrics. However, further in-depth analysis is necessary to ensure semantically meaningful results, which will be addressed in future work.

---

**Algorithm 1** Tree Training

---

**Require:** Tree degree $K$, Tree depth $d$, Network architecture $\mathcal{T}(\cdot; \boldsymbol{\theta})$, Training set $\mathcal{D} = \{(\boldsymbol{x}_i, \boldsymbol{y}_i)\}$,
     Distance measure $\ell(\cdot, \cdot)$, Loss weights $\boldsymbol{\beta} \in \mathbb{R}^{d+1}$.

1: **repeat**
     /* Sample data and infer leaves */
2:     $(\boldsymbol{x}_i, \boldsymbol{y}_i) \sim \mathcal{D}$          ▷ Sample paired training example
3:     $\{\hat{\boldsymbol{x}}_n(\boldsymbol{y}_i; \boldsymbol{\theta}), \hat{\alpha}_n(\boldsymbol{y}_i; \boldsymbol{\theta})\}_{n=1}^{K^d} \leftarrow \mathcal{T}(\boldsymbol{y}_i; \boldsymbol{\theta})$    ▷ Infer tree leaves and their probabilities

     /* Bottom-up tree construction using Eqs. (6) and (7) */
4:     Run `BottomUp`$(\{\hat{\boldsymbol{x}}_n(\boldsymbol{y}_i; \boldsymbol{\theta}), \hat{\alpha}_n(\boldsymbol{y}_i; \boldsymbol{\theta})\}_{n=1}^{K^d})$ to get:

$$\hat{\boldsymbol{x}}_{\text{MMSE}}(\boldsymbol{y}_i), \{\hat{\boldsymbol{x}}_{k_1}(\boldsymbol{y}_i), \hat{\alpha}_{k_1}(\boldsymbol{y}_i)\}_{k_1=1}^{K}, \ldots, \{\hat{\boldsymbol{x}}_{k_1,\ldots,k_d}(\boldsymbol{y}_i), \hat{\alpha}_{k_1,\ldots,k_d}(\boldsymbol{y}_i)\}_{k_1,\ldots,k_d=1}^{K}$$

     /* Accumulate $\hat{\boldsymbol{x}}$ along min path */
5:     $\hat{\mathcal{X}} \leftarrow \{\}$
6:     $\hat{\boldsymbol{x}}_{k^\star}(\boldsymbol{y}_i) \leftarrow \hat{\boldsymbol{x}}_{\text{MMSE}}(\boldsymbol{y}_i)$          ▷ Initialize closest tree node to root
7:     **repeat** $d$ **times**
8:        $\hat{\mathcal{X}} \leftarrow \hat{\mathcal{X}} \cup \hat{\boldsymbol{x}}_{k^\star}(\boldsymbol{y}_i)$          ▷ Add closest node to min path
9:        $\{\hat{\boldsymbol{x}}_k(\boldsymbol{y}_i)\}_{k=1}^{K} \leftarrow$ `GetChildren`$(\hat{\boldsymbol{x}}_{k^\star}(\boldsymbol{y}_i))$    ▷ Get $K$ children of closest node
10:       $k^\star \leftarrow \underset{k=1,\ldots,K}{\arg\min}\, \ell(\boldsymbol{x}_i, \hat{\boldsymbol{x}}_k(\boldsymbol{y}_i))$    ▷ Update closest node to closest child
11:     **end**
12:     $\hat{\mathcal{X}} \leftarrow \hat{\mathcal{X}} \cup \hat{\boldsymbol{x}}_{k^\star}(\boldsymbol{y}_i)$          ▷ Include also the closest leaf

     /* Take gradient step on weighted MSE loss to update $\boldsymbol{\theta}$ */
13:     $\mathcal{L}(\boldsymbol{\theta}) = \sum_{j=1}^{d+1} \beta_j \|\hat{\mathcal{X}}_j - \boldsymbol{x}_i\|_2^2$
14:     $\boldsymbol{\theta} \leftarrow$ `Optimizer`$(\boldsymbol{\theta}, \nabla_{\boldsymbol{\theta}} \mathcal{L})$
15: **until** converged

---

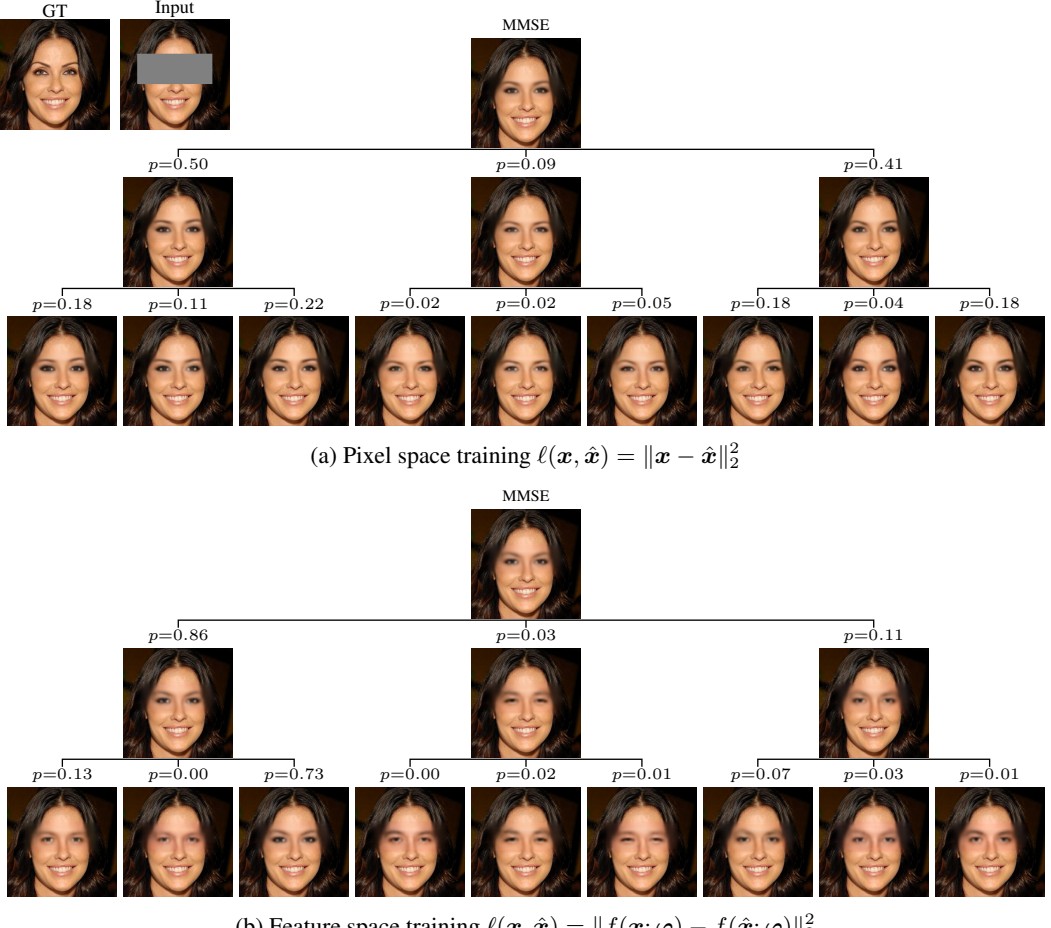

(a) Pixel space training $\ell(\boldsymbol{x}, \hat{\boldsymbol{x}}) = \|\boldsymbol{x} - \hat{\boldsymbol{x}}\|_2^2$

(b) Feature space training $\ell(\boldsymbol{x}, \hat{\boldsymbol{x}}) = \|f(\boldsymbol{x}; \boldsymbol{\varphi}) - f(\hat{\boldsymbol{x}}; \boldsymbol{\varphi})\|_2^2$

Figure A5: **Different association metrics** $\ell(\cdot, \cdot)$. (a)-(b) Predicted tree in the task of eye inpainting with an MSE association loss in pixel-space/feature-space respectively.

# E Centroidal Voronoi Tessellations

Consider a domain $\mathcal{X} \subseteq \mathbb{R}^d$ and let $\rho : \mathcal{X} \to \mathbb{R}_+$ be a weight function over this domain. Additionally, let $\{\boldsymbol{x}_i\}_{i=1}^K$ denote any set of $K$ points belonging to $\mathcal{X}$ and let $\{V_i\}_{i=1}^K$ denote any tessellation of $\mathcal{X}$ into $K$ regions. Using these notations, we can define the following loss function,

$$\mathcal{L}\left(\{(\boldsymbol{x}_i, V_i)\}_{i=1}^K\right) = \sum_{i=1}^K \int_{\boldsymbol{x} \in V_i} \rho(\boldsymbol{x}) \|\boldsymbol{x} - \boldsymbol{x}_i\|^2 \mathrm{d}\boldsymbol{x}. \tag{A7}$$

This formalism arises in various applications such as data compression, optimal quantization, clustering, and other applications [66]. In all of these applications, our goal is the same; we want to minimize Eq. (A7) to obtain optimal performance.

A fundamental result in this setting is that for Eq. (A7) to be minimized, it is necessary that $\{V_i\}_{i=1}^K$ are the Voronoi regions corresponding to $\{\boldsymbol{x}_i\}_{i=1}^K$ and, simultaneously, $\{\boldsymbol{x}_i\}_{i=1}^K$ are the *centroids* of the corresponding $\{V_i\}_{i=1}^K$. Namely, $\{(\boldsymbol{x}_i, V_i)\}_{i=1}^K$ form a special tessellation of space, commonly referred to as a Centroidal Voronoi Tessellation (CVT). Specifically, it can be shown that the optimal centroids of the resulting CVT are given by

$$\boldsymbol{x}_i = \frac{\int_{\boldsymbol{x} \in V_i} \boldsymbol{x} \rho(\boldsymbol{x}) \mathrm{d}\boldsymbol{x}}{\int_{\boldsymbol{x} \in V_i} \rho(\boldsymbol{x}) \mathrm{d}\boldsymbol{x}}, \tag{A8}$$

where $\{V_i\}_{i=1}^K$ form a Voronoi tessellation of $\mathcal{X}$ w.r.t. $\{\boldsymbol{x}_i\}_{i=1}^K$, and are defined as

$$V_i = \{\boldsymbol{x} : \|\boldsymbol{x} - \boldsymbol{x}_i\|_2 < \|\boldsymbol{x} - \boldsymbol{x}_j\|_2, \quad \forall j \neq i\}. \tag{A9}$$

In general, the solution to Eq. (A7) is not unique, and no existing algorithm is guaranteed to converge to an optimal solution. Nevertheless, efficient algorithms like Lloyd's algorithm (Voronoi Iteration) are known to converge to a local minimum, which frequently produces satisfactory results in practice.

# F   Unconditional Hierarchical $K$-means

In Section 3.1 we mentioned that in a discrete setting, our approach is equivalent to a *hierarchical* $K$-means clustering. Here, we describe the hierarchical $K$-means algorithm and review its properties in the standard (unconditional) setting.

Given a set of $N$ data points $\{\boldsymbol{x}_i\}$ where $\boldsymbol{x}_i \in \mathbb{R}^{d_x}$, the goal of $K$-means is to partition the data points into $K$ clusters/sets $\{\mathcal{C}_1, \ldots, \mathcal{C}_K\}$, such that each data point belongs to the cluster with the nearest mean/centroid serving as a prototype of the cluster. This results in a partitioning/tessellation of the data space into centroidal Voronoi cells, where for each cluster the within-cluster variances (squared Euclidean distances) are minimized. Formally, the objective function for finding the clusters is given by

$$\arg\min_{\mathcal{C}_1,\ldots,\mathcal{C}_K} \sum_{k=1}^{K} \sum_{\boldsymbol{x}_i \in \mathcal{C}_k} \|\boldsymbol{x}_i - \boldsymbol{\mu}_k\|_2^2, \tag{A10}$$

where $\boldsymbol{\mu}_k$ is the mean/centroid of cluster $\mathcal{C}_k$, given by $\boldsymbol{\mu}_k = \frac{1}{|\mathcal{C}_k|} \sum_{\boldsymbol{x}_i \in \mathcal{C}_k} \boldsymbol{x}_i$. While in general Problem (A10) is NP-hard, given some initial means/centroids, the $K$-means algorithm (also known as Lloyd's method) finds a local minimum by alternating between (i) assigning each data point to the cluster with the nearest mean, and (ii) updating the cluster means based on the new assignment.

Extending $K$-means hierarchically is rather straightforward. Starting from a single cluster comprised of all data points $\{\boldsymbol{x}_i\}$, at each level of the hierarchy, the data points belonging to the $k$th cluster $\mathcal{C}_k$ are split further into $K$ sub-clusters $\{\mathcal{C}_{k,1}, \ldots, \mathcal{C}_{k,K}\}$ by applying $K$-means. We refer to this extended algorithm as *hierarchical $K$-means*. The result of applying hierarchical $K$-means $d$ times is a dendrogram/top-down balanced tree $\mathcal{T}$ of degree $K$, depth $d$, and breadth (final number of leaves) $K^d$. Note that this successive process is different from applying $K$-means with $K^d$ clusters once. This is because the hierarchical memberships are enforced, restricting data points in subclusters $\{\mathcal{C}_{i,1}, \ldots, \mathcal{C}_{i,K}\}$ from being assigned to subclusters $\{\mathcal{C}_{j,1}, \ldots, \mathcal{C}_{j,K}\}$ if $i \neq j$.

Hierarchical $K$-means enjoys several advantages compared to the standard ("Flat") $K$-means algorithm. One of these advantages is increased robustness to initialization as demonstrated in Appendix G.3.

# G Toy Example

## G.1 Model Architecture

The model used to produce our results in Fig. 3 consisted of two 5-layer MLPs with 256 hidden units, one for predicting the leaves and one for predicting the likelihoods. Both MLPs consisted of linear layers interleaved with the SiLU activation. In addition, the output of the second MLP was passed through a $\mathrm{softmax}$ to represent a valid probability distribution.

## G.2 Analytical Posterior

In Fig. 3, we plotted the posterior trees on the (unknown) ground truth posterior. Here we provide the closed-form expression for the analytically derived posterior for completeness. The denoising task we assumed was $\mathbf{y} = \mathbf{x} + \mathbf{n}$, where $\mathbf{x}$ comes from a mixture of $L = 4$ Gaussians, $p_{\mathbf{x}}(\boldsymbol{x}) = \sum_{\ell=1}^{L} \pi_\ell \mathcal{N}(\boldsymbol{x}; \boldsymbol{\mu}_\ell, \boldsymbol{\Sigma}_\ell)$, and $\mathbf{n} \sim \mathcal{N}(\cdot; \mathbf{0}, \sigma_\varepsilon^2 \boldsymbol{I})$ is a white Gaussian noise. Specifically, in our toy example we used equally probable spherical Gaussians (*i.e.*, $\pi_\ell = \frac{1}{4}$, $\boldsymbol{\Sigma}_\ell = \boldsymbol{I}$), with the following means

$$\boldsymbol{\mu}_1 = \begin{pmatrix} -6.0 \\ +2.5 \end{pmatrix}, \; \boldsymbol{\mu}_2 = \begin{pmatrix} +1.0 \\ +2.5 \end{pmatrix}, \; \boldsymbol{\mu}_3 = \begin{pmatrix} -2.5 \\ +6.0 \end{pmatrix}, \; \boldsymbol{\mu}_4 = \begin{pmatrix} -2.5 \\ -1.5 \end{pmatrix}. \tag{A11}$$

Let c be an auxiliary random variable taking values in $\{1, \ldots, L\}$ with probabilities $\{\pi_1, \ldots, \pi_L\}$. Then we can write the posterior by invoking the law of total probability conditioned on the event $\mathrm{c} = \ell$,

$$\begin{aligned} p(\boldsymbol{x}|\boldsymbol{y}) &= \sum_{\ell=1}^{L} p_{\mathbf{x}|\mathbf{y},\mathrm{c}}(\boldsymbol{x}|\boldsymbol{y}, \ell) p_{\mathrm{c}|\mathbf{y}}(\ell|\boldsymbol{y}) \\ &= \sum_{\ell=1}^{L} p_{\mathbf{x}|\mathbf{y},\mathrm{c}}(\boldsymbol{x}|\boldsymbol{y}, \ell) \frac{p_{\mathbf{y}|\mathrm{c}}(\boldsymbol{y}|\ell) p_{\mathrm{c}}(\ell)}{p_{\mathbf{y}}(\boldsymbol{y})} \\ &= \sum_{\ell=1}^{L} \mathcal{N}(\boldsymbol{x}; \tilde{\boldsymbol{\mu}}_\ell, \tilde{\boldsymbol{\Sigma}}_\ell) \frac{q_\ell \pi_\ell}{\sum_{\ell'=1}^{L} q_{\ell'} \pi_{\ell'}}, \end{aligned} \tag{A12}$$

where the first step is by Bayes rule, and in the result we denoted

$$\begin{aligned} q_\ell &= \mathcal{N}(\boldsymbol{y}; \boldsymbol{\mu}_\ell, \boldsymbol{\Sigma}_\ell + \sigma_\varepsilon^2 \boldsymbol{I}), \\ \tilde{\boldsymbol{\mu}}_\ell &= \boldsymbol{\mu}_\ell + \boldsymbol{\Sigma}_\ell (\boldsymbol{\Sigma}_\ell + \sigma_\varepsilon^2 \boldsymbol{I})^{-1}(\boldsymbol{y} - \boldsymbol{\mu}_\ell) \\ \tilde{\boldsymbol{\Sigma}}_\ell &= \boldsymbol{\Sigma}_\ell - \boldsymbol{\Sigma}_\ell (\boldsymbol{\Sigma}_\ell + \sigma_\varepsilon^2 \boldsymbol{I})^{-1} \boldsymbol{\Sigma}_\ell, \quad \ell = 1, \ldots, L. \end{aligned} \tag{A13}$$

In Fig. A6, in a similar fashion to Fig. 3 from the main text, we visualize posterior trees for additional test inputs $\boldsymbol{y}_t$. As clearly evident in all cases, our method recovers the (approximated) ground truth posterior tree with high accuracy.

## G.3 Stability of "Flat" vs. Hierarchical $K$-means

As mentioned earlier, the hierarchical $K$-means algorithm has several advantages over the classical "Flat" $K$-means algorithm. Here, we examine the effect of the random initialization on the resulting clusters found by "Flat"/Hierarchical K-means. In both cases, we applied the respective method to 10K posterior samples to avoid errors resulting from an insufficient sample size. Figure A7 demonstrates the resulting clusters for 3 different seeds. As can be seen by the result, despite using the widely adopted $K$-means++ initialization for both algorithms, "Flat" $K$-means was less resilient to a bad initialization compared to its hierarchical counterpart. This is yet another advantage of working with trees of depth bigger than $d = 1$.

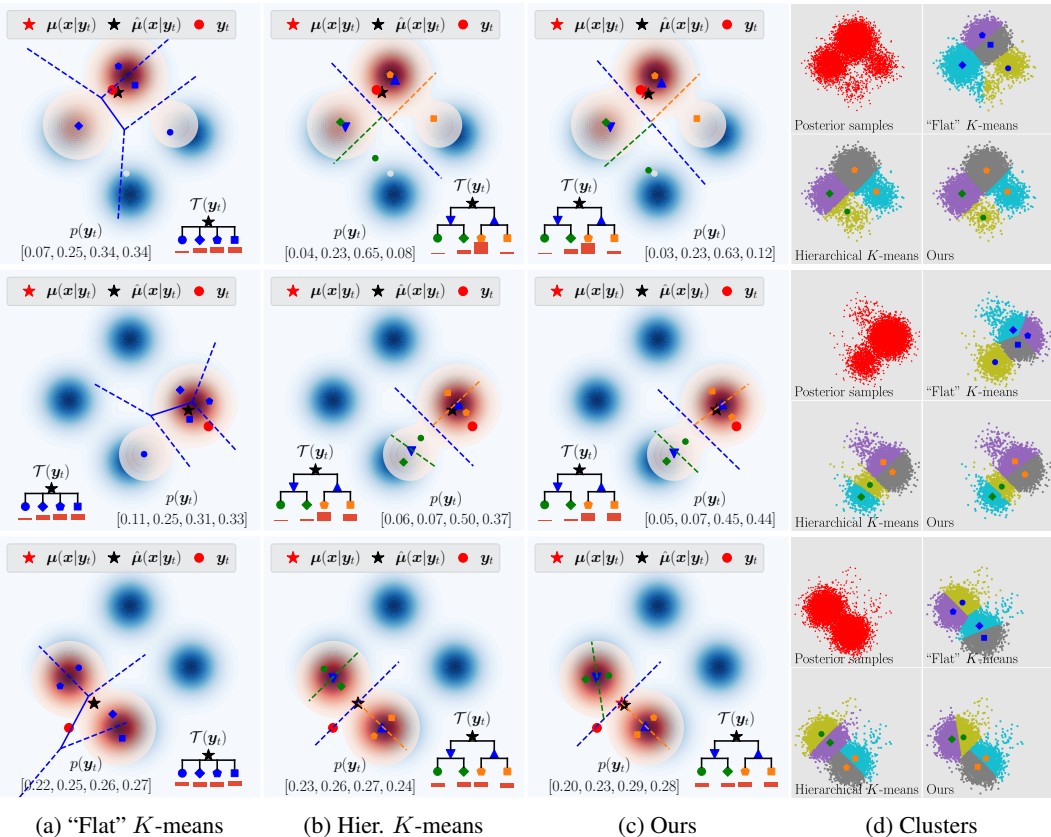

(a) "Flat" $K$-means     (b) Hier. $K$-means     (c) Ours     (d) Clusters

Figure A6: **Additional test points** $\boldsymbol{y}_t$. Each row above shows the results for a different test point $\boldsymbol{y}_t$ (red dot) when clustering with (a) "Flat" $K$-means, (b) Hierarchical $K$-means, (c) Posterior trees (ours). In (d) we show the resulting partition/clustering induced by each method by coloring 10K samples from the posterior $p_{\mathbf{x}|\mathbf{y}}(\boldsymbol{x}|\boldsymbol{y}_t)$ according to their nearest cluster. Our method consistently recovers the result of hierarchical $K$-means in all cases. Moreover, other than trivial cases such as the bottom row, the hierarchy better represents weaker modes compared to "Flat" $K$-means.

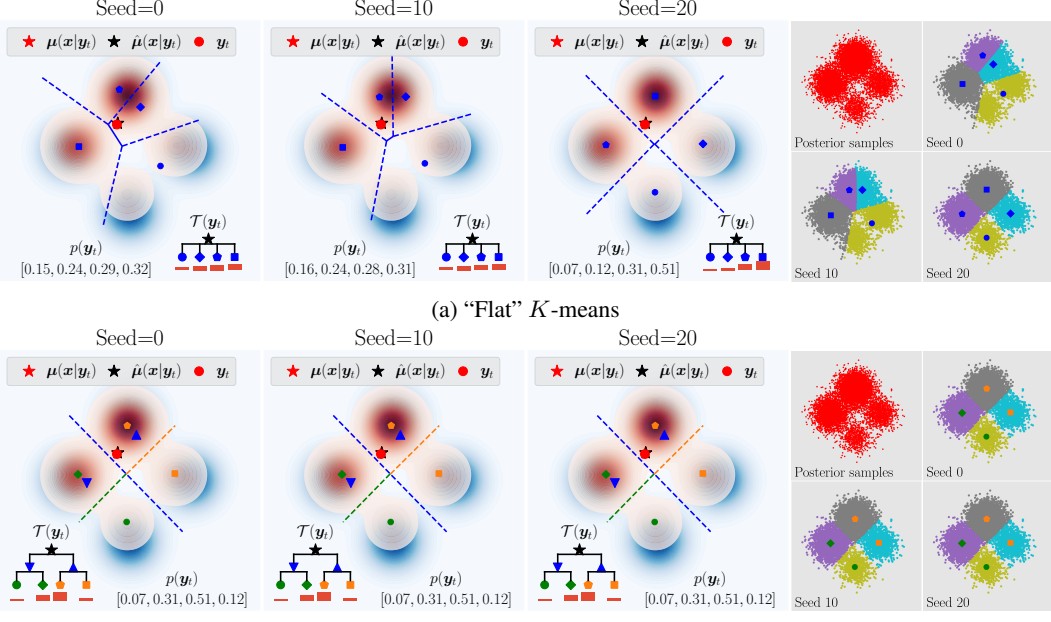

(a) "Flat" $K$-means

(b) Hierarchical $K$-means

Figure A7: **Resilience to initialization**. Here we inspect the effect of the seed used for initialization, on the resulting clusters recovered by (a) "Flat" $K$-means and (b) Hierarchical $K$-means. The hierarchy results in a more resilient algorithm that recovers the same result regardless of the seed used for initialization.

# H   Tree Width vs. Depth

Throughout Section 4 we presented results with trees of degree $K = 3$ and depth $d = 2$. This was done only for the convenience of the exposition and our method is not restricted to this specific layout. Here we examine the effect of tree width compared to tree depth. Figures A8 and A9 presents the resulting trees for four different model configurations on the task of digit inpainting. For a fair comparison between model pairs, we fix the number of output leaves which dictates the number of model parameters, and only change the hierarchical structure. Figure A8(a) compares a layout of $K = 4, d = 1$ to $K = 2, d = 2$. As evident in the result, the hierarchy better organizes the different options, presenting the "4" cluster more faithfully than the flat tree which presents a cluster midway between a "4" and a "1". Figure A9 repeats this experiment for deeper trees comparing $K = 4, d = 2$ to $K = 2, d = 4$. A couple of remarks are in order regarding the results. First, in both cases, the "4" cluster has a similar probability relative to the "1" cluster. This indicates that our method is consistent in the predicted posterior modes and their likelihoods, with the only change between different layouts being the chosen tessellation of the output space. Second, the degree $K$ controls the emphasis/over-representation devoted to weaker posterior modes. A smaller degree leads to more emphasis on weaker modes as tree depth grows. Lastly, it is important to note that the optimal layout $K$ and $d$ is task and input-dependant, and setting these adaptively is an interesting direction for future research.

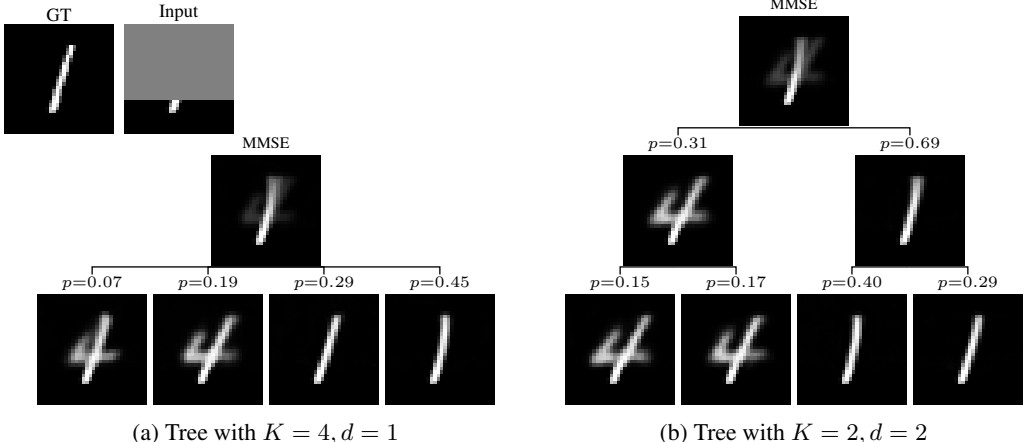

(a) Tree with $K = 4, d = 1$        (b) Tree with $K = 2, d = 2$

Figure A8: **Flat vs. hierarchical trees.** (a) Flat tree with $K = 4, d = 1$. (b) Binary tree with $K = 2, d = 2$, with overall 4 leaves. The binary tree categorizes the leaves and better represents the "4" mode.

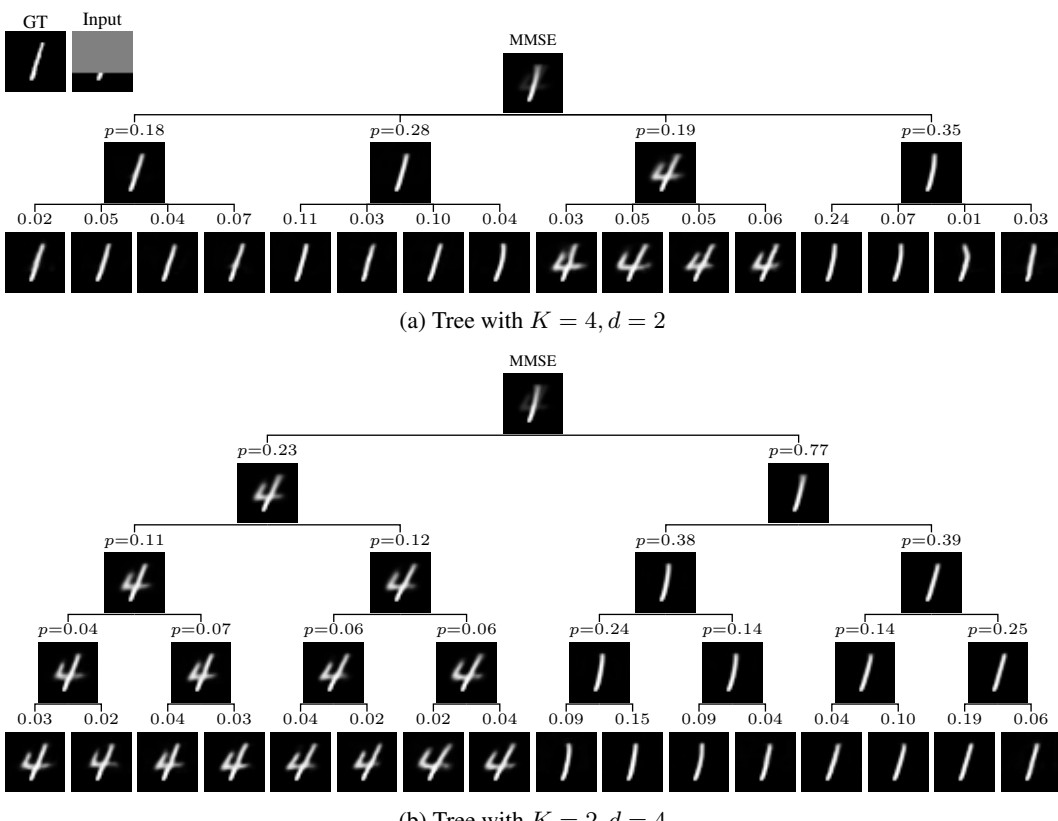

(a) Tree with $K = 4, d = 2$

(b) Tree with $K = 2, d = 4$

Figure A9: **Tree depth vs. width.** (a) Tree with $K = 4, d = 2$. (b) Binary tree with $K = 2, d = 4$ also resulting in 16 leaves. The binary tree emphasizes more the "4" mode relative to the "1" mode, although in both cases the probability mass associated with the "4" is $\approx 20\%$.

# I GAN-based Posterior Samplers

In Section 4 we compared our method mainly to diffusion-based posterior samplers (*e.g.,* [7, 8, 11]). While these samplers often lead to state-of-the-art sample quality, they are known to be computationally intensive. In theory, GAN-based samplers such as MAT [10], can significantly speed up the proposed two-step baseline of sampling followed by hierarchical $K$-means; however, as evident in Fig. A10, even top-performing GAN-based samplers often collapse to a single mode of the posterior, producing samples with little variability that do not faithfully reflect the full distribution.

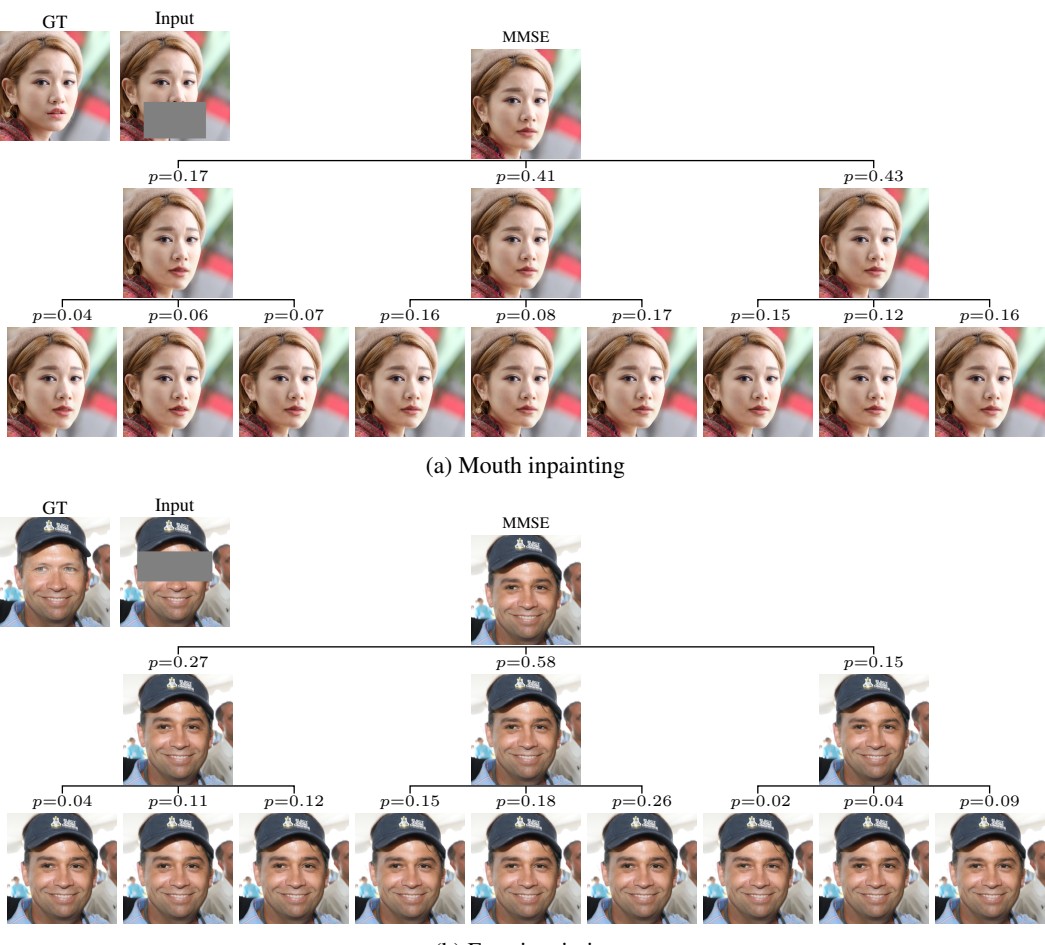

(a) Mouth inpainting

(b) Eyes inpainting

Figure A10: **GAN-based posterior sampler.** (a)-(b) Posterior trees constructed with MAT, by applying hierarchical $K$-means to 100 samples in the tasks of mouth/eyes inpainting respectively. As evident in both cases, the resulting tree exhibits little to no variance due to mode collapse.

## J  Visual Comparison to Baselines

In Table 1 we reported quantitative comparisons to the proposed baseline implemented with various samplers. In Figs. A11 and A12, we include two sample visual comparisons of the resulting trees with our method and the baselines for the tasks of face colorization and mouth inpainting.

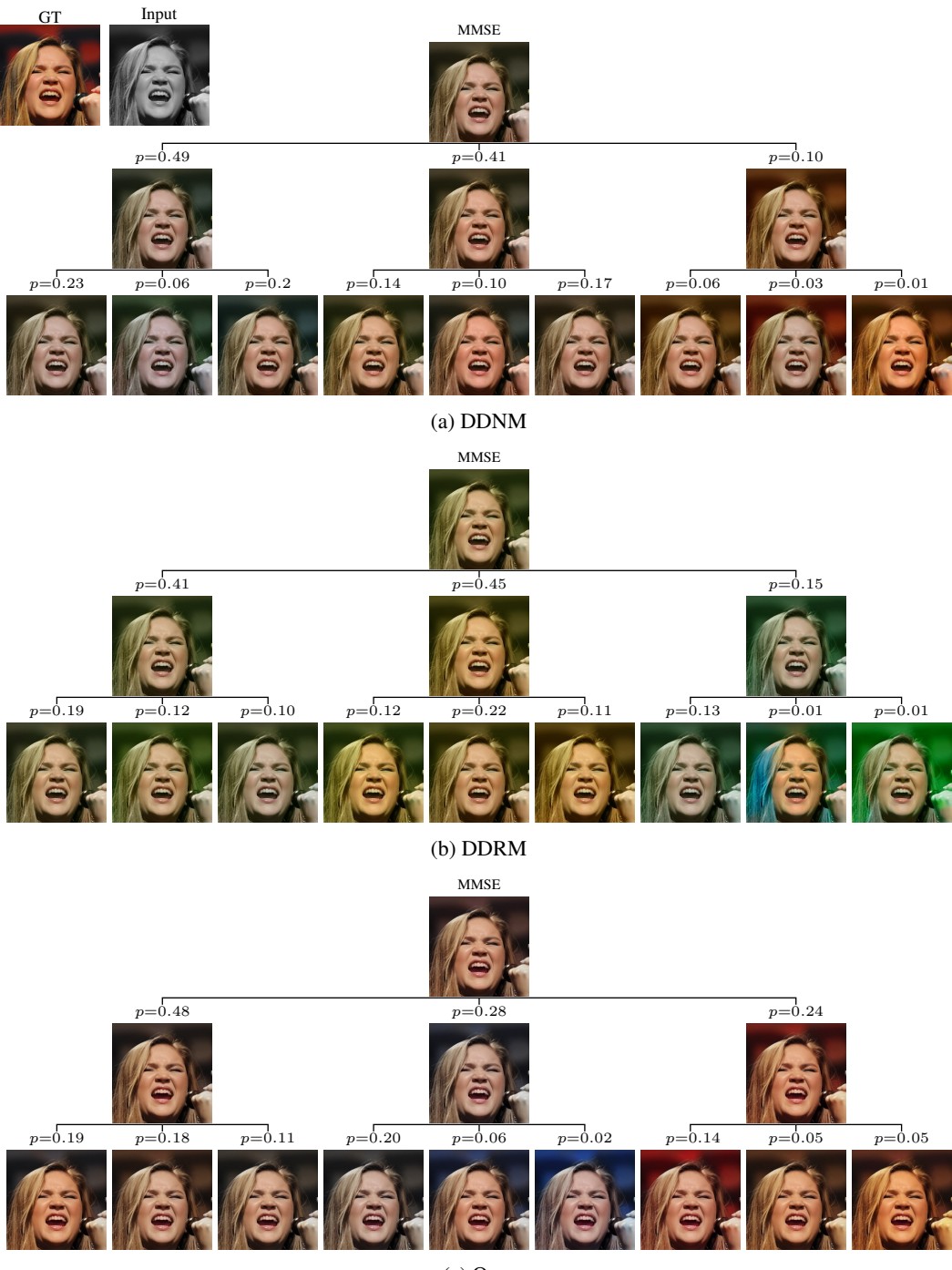

Figure A11: **Tree comparison in colorization.**

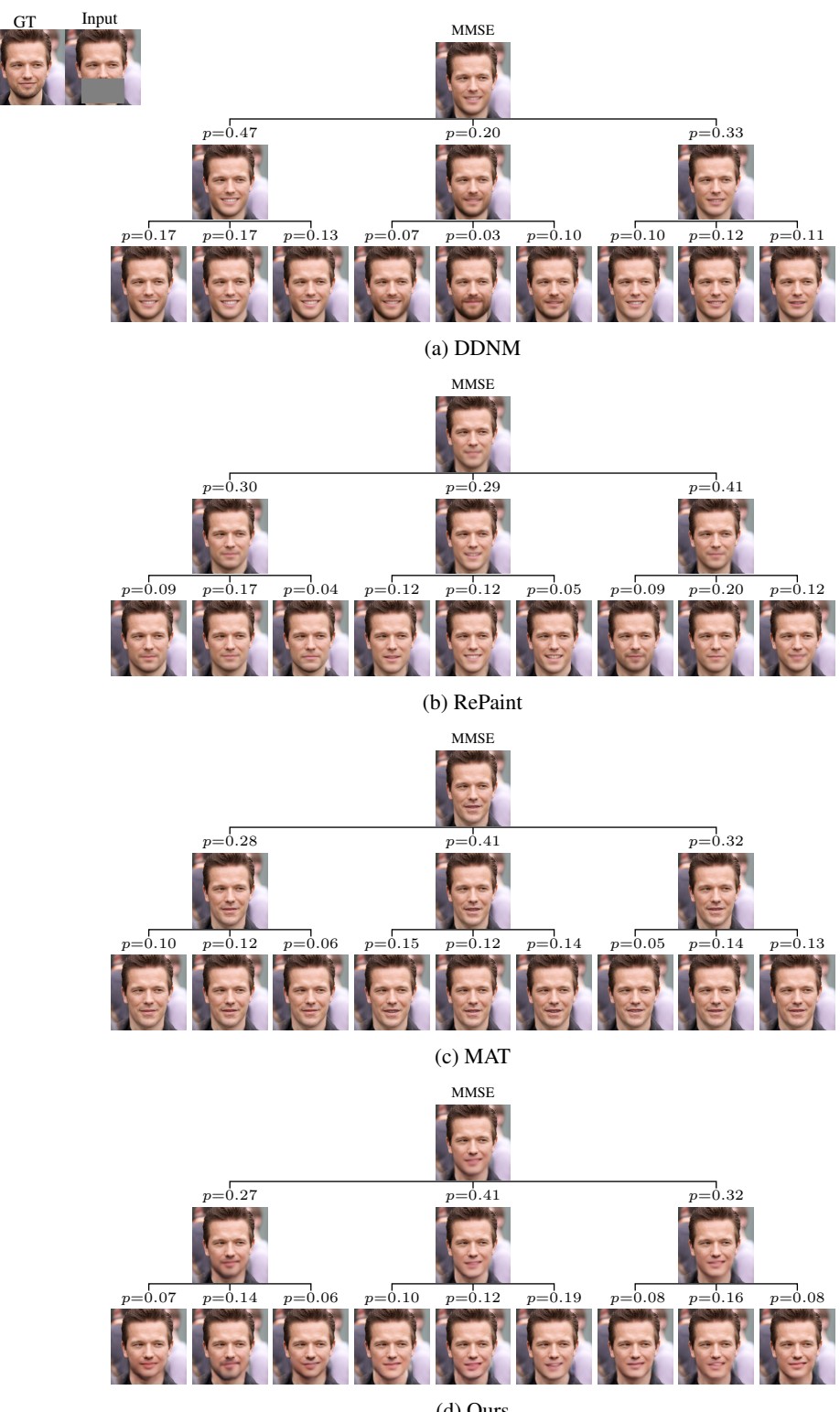

(a) DDNM

(b) RePaint

(c) MAT

(d) Ours

Figure A12: **Tree comparison in mouth inpainting.**

# K   Runtime Comparison as a Function of Batch Size

In Table 1 we reported the runtime and memory footprint assuming a batch of size 1. However, in GPU computing, memory footprint and hardware parallelization ultimately determine the overall runtime. We tried to refrain from factoring these in our calculation in Table 1 as these numbers are hardware-specific depending on the available GPU. Nonetheless, it is important to note that our method is not only faster than the competing approaches with a batch of size 1, it also has a lower memory footprint and is therefore more amenable to parallelization. Table 2 benchmarks the speed of the forward pass and the GPU memory usage as a function of the batch size on an A6000 GPU with 48 GB. For each method, we tested three different batch sizes: 1, 10, and the maximal number of samples that fit in memory. The forward pass speed and the memory usage in each setting were tested 100 times using CUDA events (reported as mean $\pm$ std). Our method is extremely fast and parallelizable, enabling the inference of 430 test images with a single forward pass of 1.15 seconds.

Even in the case of a **single** test image, our method is far superior. For simplicity let us assume the cost of running hierarchical $K$-means is negligible, and that we can squeeze a batch size of $\approx 100$ samples for the diffusion-based samplers, and $\approx$"33.33" samples for MAT. Sampling $N_s = 100$ posterior samples with DDRM (fastest diffusion baseline, requires only 20 denoising steps) lasts 46.3 seconds, and with MAT lasts 7.8 seconds. In comparison, our method requires only 0.014 seconds, leading to a $557\times$ speedup compared to the fastest baseline.

Table 2: Runtime comparison as a function of batch size on an Nvidia A6000 GPU. Note that unlike Table 1, here we report the speed of a forward pass separately from the number of required forward passes to infer a posterior tree for a single test image.

| Method | Batch Size | Forward pass speed (ms) | Memory (GB) | Forward passes per tree | Total time per tree (sec) |
|---|---|---|---|---|---|
| Diffusion | 1 | 34$\pm$0.4 | 1.85 | 2,000-456,000 | 68-15,504 |
| | 10 | 24.5$\pm$0.1 | 7.5 | | 49-11,172 |
| | 97 | 23.9$\pm$0.6 | 46.8 | | 48-10,898 |
| MAT | 1 | 150$\pm$0.8 | 3.0 | 100 | 15 |
| | 10 | 86.7$\pm$1.2 | 16.0 | | 8.67 |
| | 30 | 86.6$\pm$2.5 | 47.2 | | 8.66 |
| Ours | 1 | 14$\pm$0.2 | 1.3 | 1 | 0.014 |
| | 10 | 3.9$\pm$0.1 | 2.1 | | 0.0039 |
| | 430 | 2.67$\pm$0.1 | 46.9 | | 0.00267 |

## L   Baselines Performance as a Function of Compute

As mentioned in Section 4.1, baseline trees were computed using a two-step procedure: (1) Sampling $N_s$ images from the posterior, and (2) performing hierarchical $K$-means $d$ times to build a tree of degree $K$ and depth $d$. Therefore, saving computation in this process requires sampling fewer images $N_s$ per test input. However, in our experiments with $K = 3, d = 2$ (i.e. a tree with 9 leaves), we noticed that using less than $N_s = 100$ images often led to degenerate trees with one or more leaves having 1 sample or less. For example, consider the case of using $N_s = 9$ samples. Even if the posterior is perfectly balanced (often not the case), each leaf will have only 1 sample to "average" at depth 2. Figure A13 plots the success probability of building a tree with $K^d = 9$ leaves as a function of the number of sampled images $N_s$ in the different tested tasks. A tree is considered to be successfully constructed if each of its leaves has at least 2 samples to average. As evident in the results (*e.g.,* Fig. A13(b)), it is likely that to achieve the optimal performance with the baselines more than $N_s = 100$ samples are needed, which would be even more computationally demanding. Moreover, the number of samples needed to successfully construct trees with more than 9 leaves is expected to be significantly larger rendering this baseline impractical, especially when considering a sizable test set.

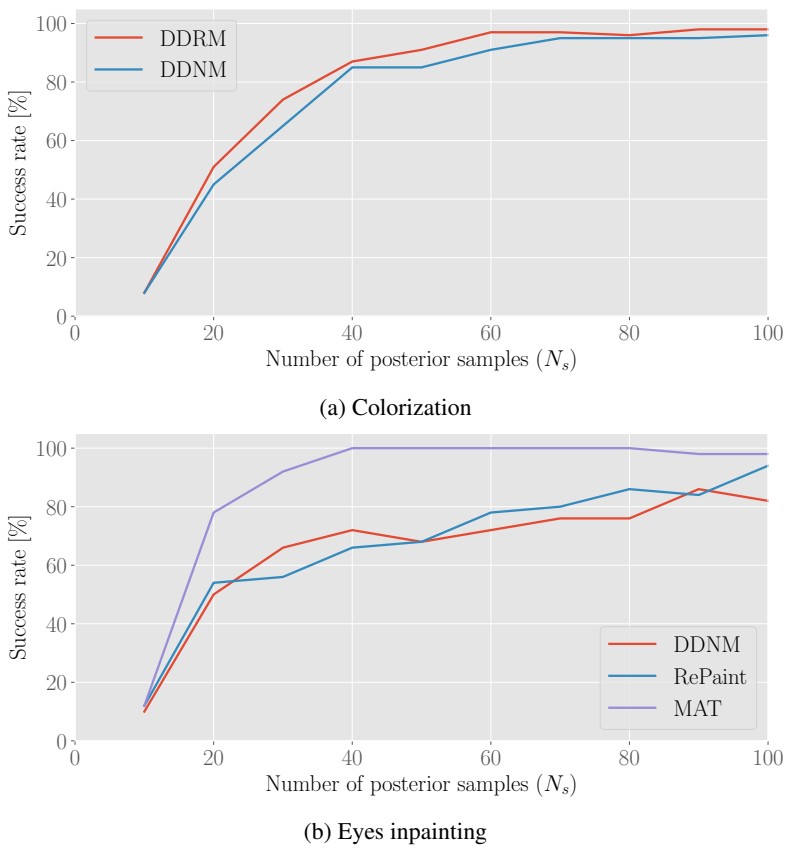

(a) Colorization

(b) Eyes inpainting

Figure A13: **Tree success rate vs. number of posterior samples** $N_s$**.** (a)-(b) Success rate of building a (posterior) tree by applying hierarchical $K$-means to $N_s$ samples in the tasks of colorization and eyes inpainting respectively.

# M More Results

Here we include more results for each of the tasks presented in Section 4 from the main text. Additionally, we also include colorization results on the AFHQ(v2) dataset [67], with a similar performance to that obtained on CelebA-HQ.

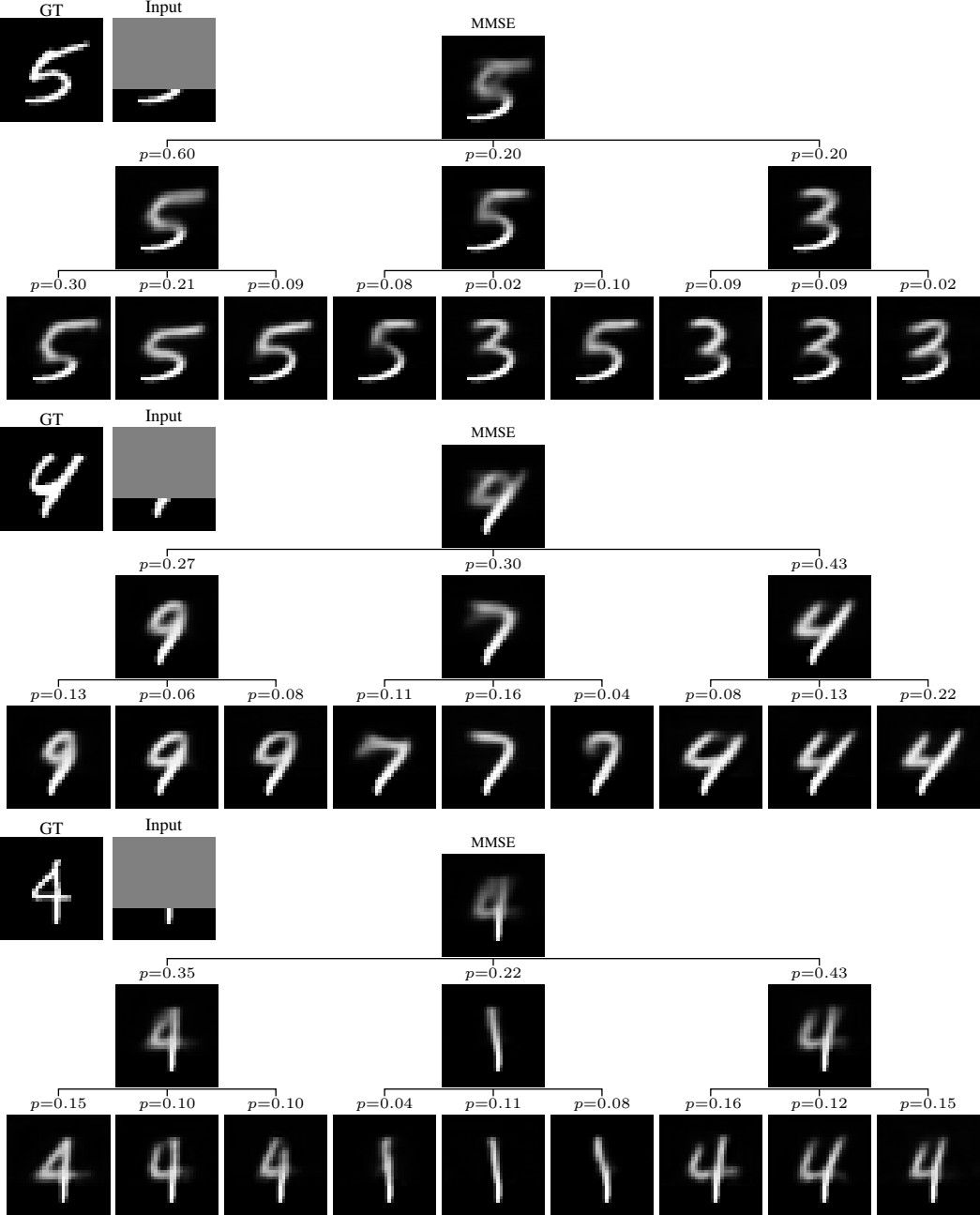

Figure A14: **More digit inpainting results.**

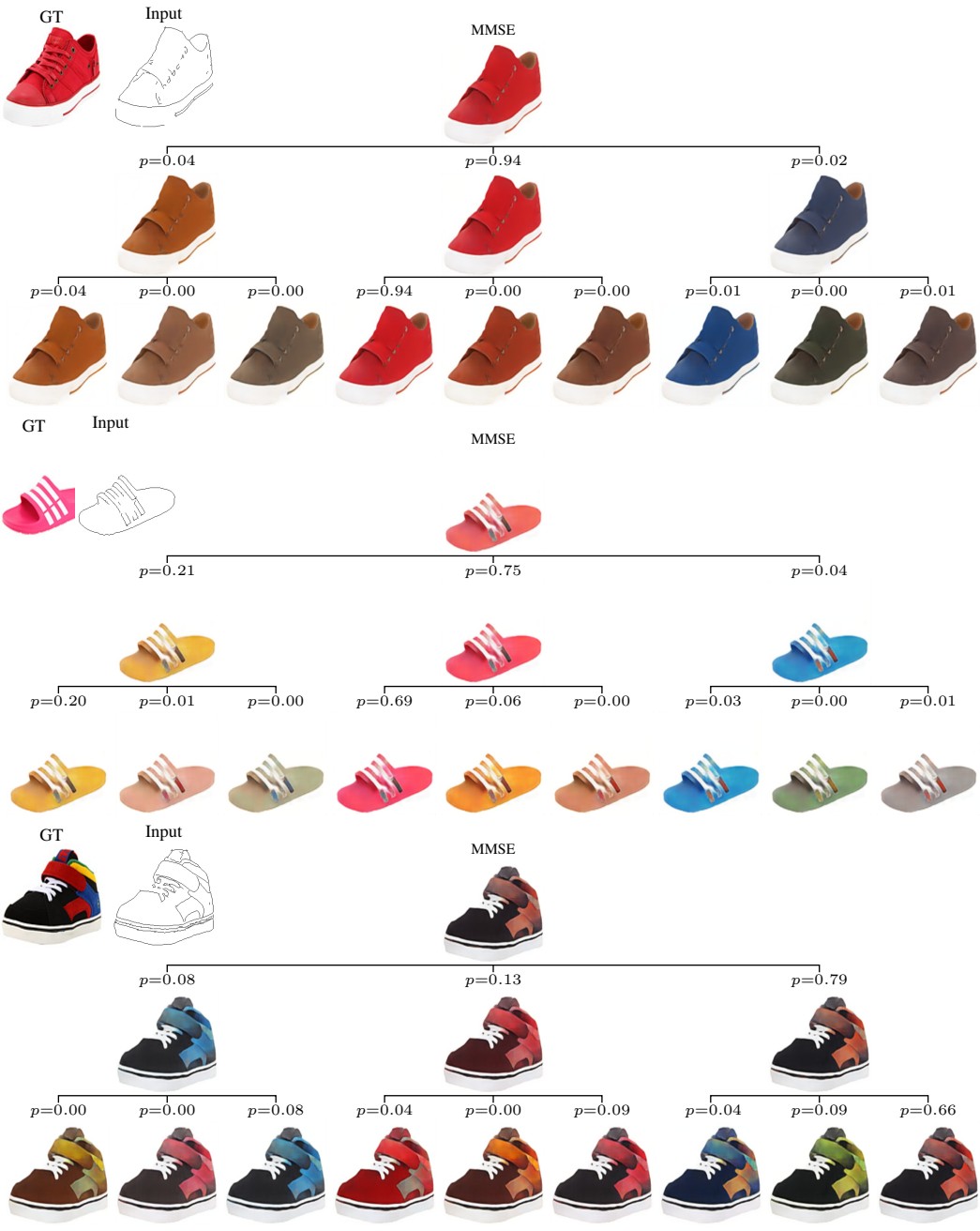

Figure A15: **More Edges→Shoes results.**

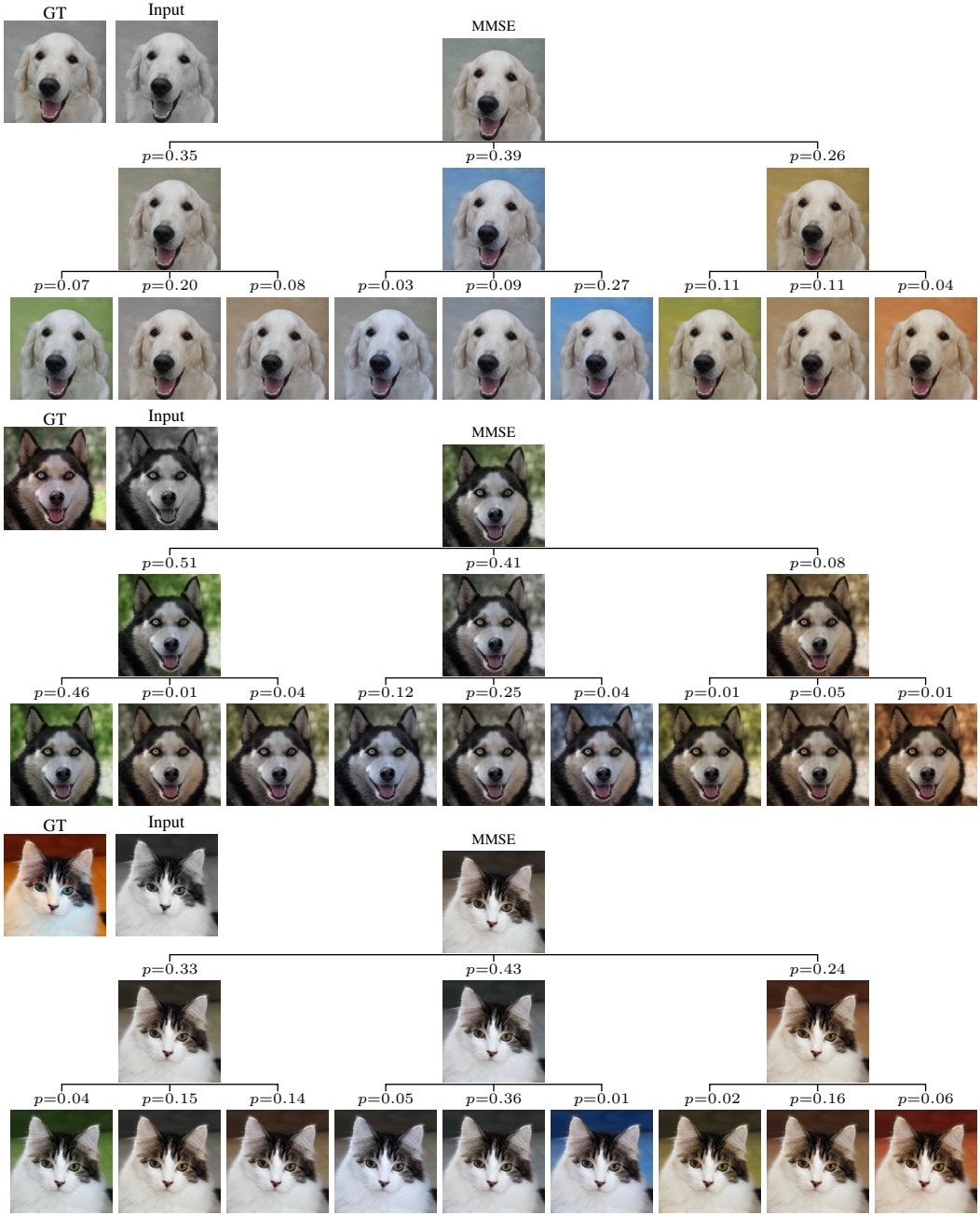

Figure A16: **AFHQ(v2) colorization results.**

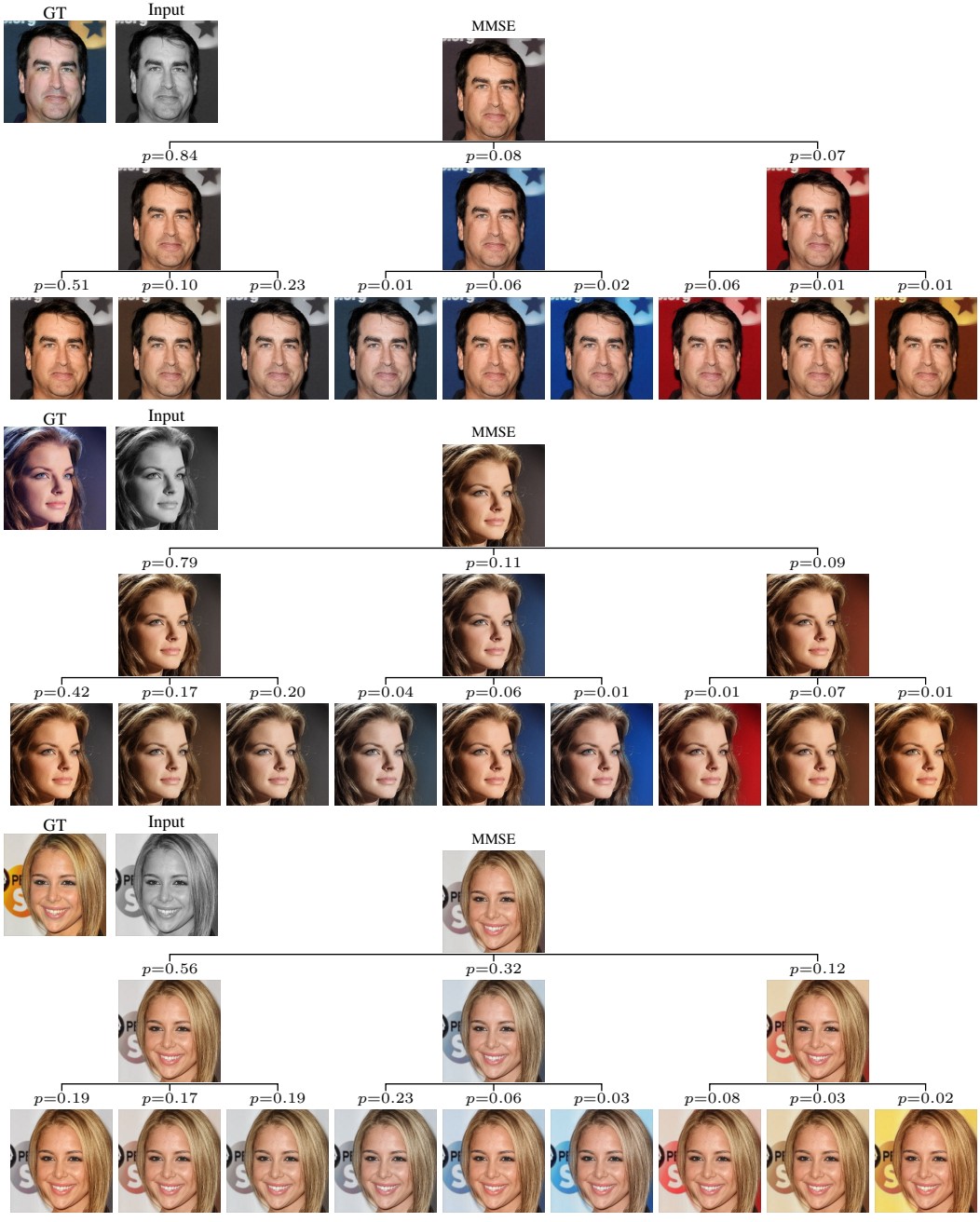

Figure A17: **More CelebA-HQ colorization results.**

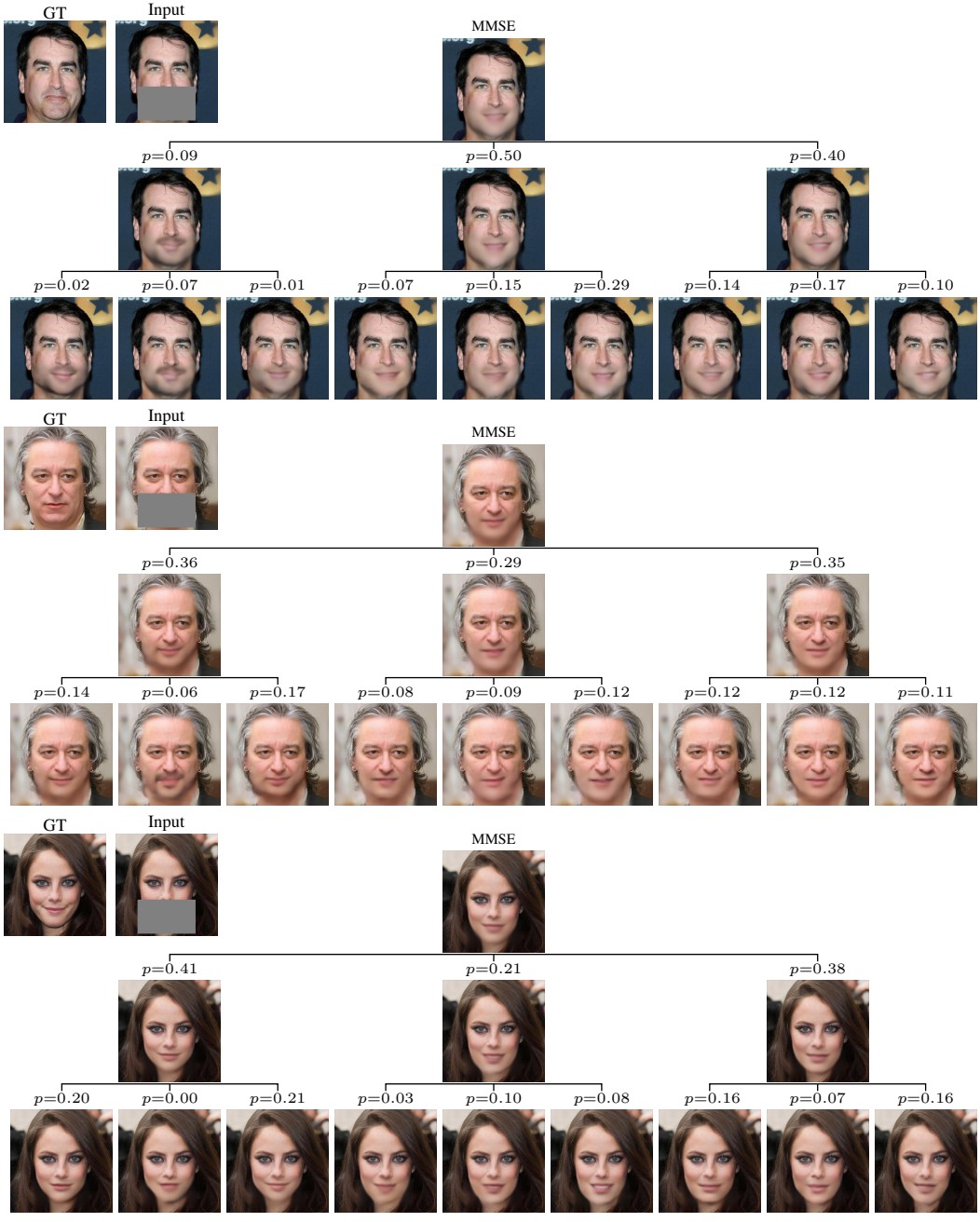

Figure A18: **More mouth inpainting results.**

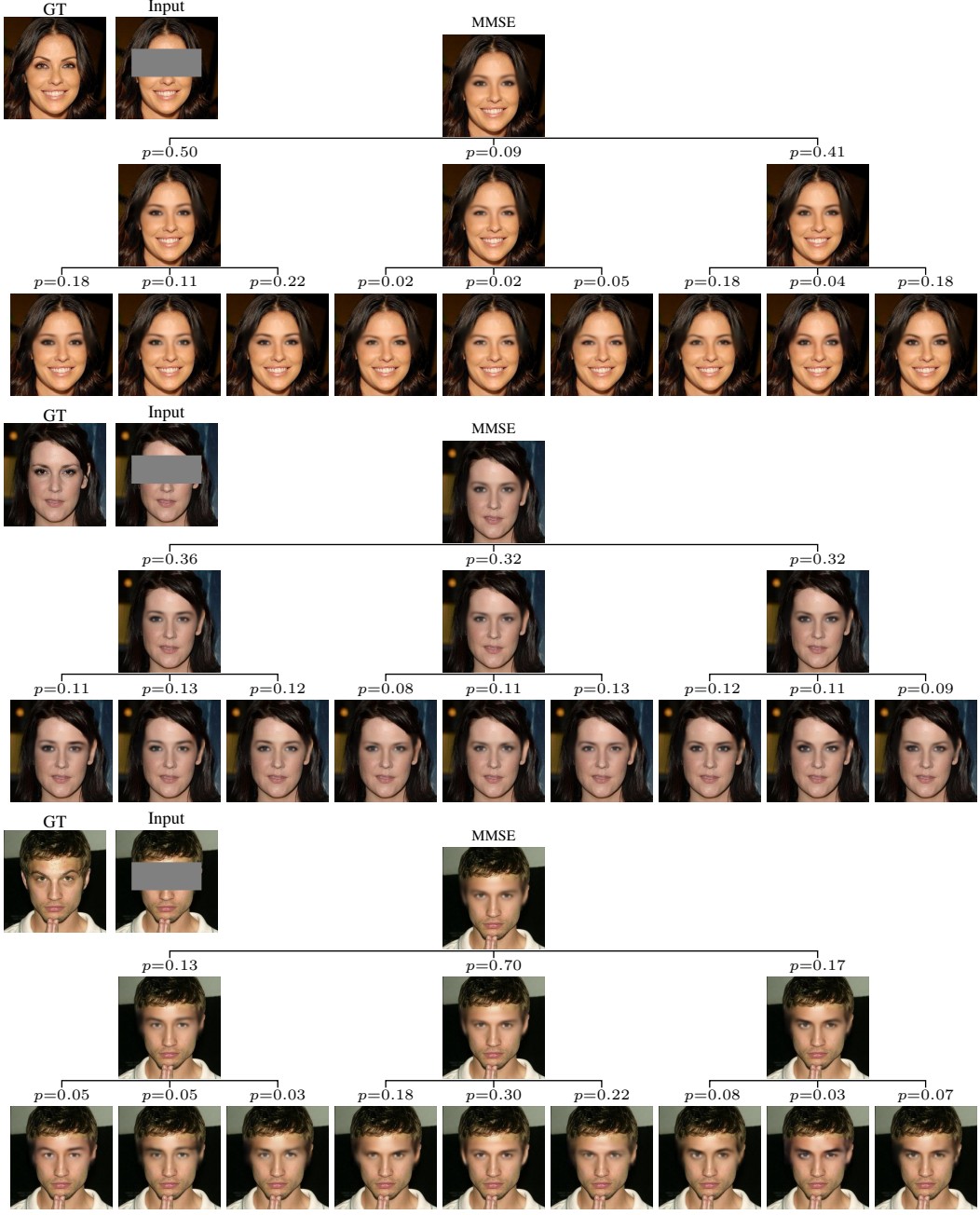

Figure A19: **More eyes inpainting results.**

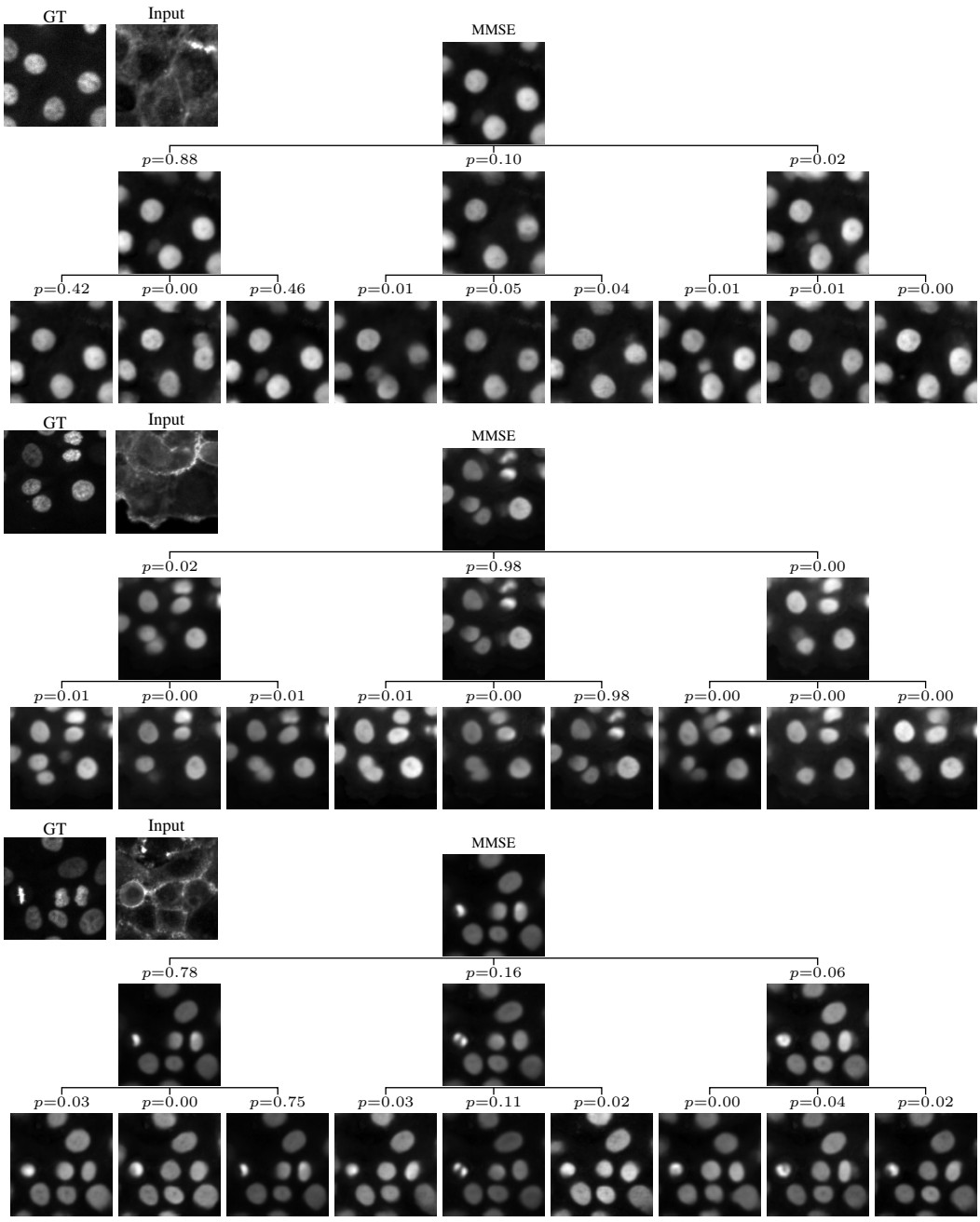

Figure A20: **More Bioimage translation results.**

