# OpenReview forum: "Hierarchical Uncertainty Exploration via Feedforward Posterior Trees"
_NeurIPS.cc/2024/Conference — NeurIPS 2024 poster_

### Official Review · Reviewer_VHHJ · 2024-07-11

**Soundness:** 3
**Presentation:** 3
**Contribution:** 2
**Rating:** 7
**Confidence:** 3

**Summary:**

The paper proposes a method to obtain a hierarchical representation of samples from the posterior distribution of inverse problems. This method can be integrated into any existing approach to learn and sample from the posterior distribution. It involves learning a tree structure where each node represents a prototype reconstruction, with child nodes refining their parent node. This allows for the exploration of different modes and levels of detail in the solution space.

**Strengths:**

**Originality.** While the method has connections to post-hoc hierarchical clustering of samples, it is novel in that it allows for faster visualization of the posterior distribution samples at test time.

**Quality and clarity.** Overall, the paper is well-structured, easy to follow, and sufficiently motivates the problem. However, the clarity of the presentation of the proposed loss function could be improved.

**Significance.** The problem of informatively visualizing the variability in samples from the posterior distribution of inverse problems is impactful.

**Weaknesses:**

* While I truly appreciate the importance of informatively visualizing samples from the posterior distribution, the proposed method seems to require several heuristics for training to yield desirable results (e.g., discussions in sections 3.3 and 3.4). Are the computational complexity gains of the proposed method, compared to performing post-hoc hierarchical clustering, worth the need for tuning the parameters in these heuristics?

* It seems like the architecture of the network required to predict the tree depends on the tree depth and width as $K^d$, where $d$ is the tree depth and $K$ is the branching factor. This appears to be a very limiting factor in the depth and width of the tree that can be learned. How does the cost of learning the tree structure as we scale $K$ and $d$ compare to the cost of post-hoc hierarchical clustering?

**Questions:**

* When augmenting existing methods for learning conditional generative models (for learning the posterior), how does the loss associated with the tree structure affect the learning of the generative model? How do the authors prevent the proposed loss from negatively biasing the learning of the posterior distribution?

* What is the computational overhead of the proposed method compared to simply training a generative model to learn the posterior distribution? What is the cost of post-hoc hierarchical clustering? How many inverse problems need to be solved to justify the extra training time?

**Limitations:**

Authors provide a discussion on the limitations of their
method.

---

> ### Author Rebuttal · Authors · 2024-08-06
>
> **Tree loss function and training scheme**
> Thanks for pointing out this is not clear enough. We will include an appendix with the explicit training algorithm to better clarify the loss function and our overall training scheme. Kindly note that we trained our models from scratch on a standard dataset of a **single** posterior sample per input, and did not augment existing conditional generative models. After training with a hierarchical MCL loss, for every input, our model learns to predict a tree that hierarchically discretizes the posterior distribution. In our experiments, the various posterior sampling baselines are only used in the comparisons to benchmark our results and quantify the quality of the predicted trees.
>
> **Runtime comparisons**
> Table 1 compares the speed of our method and the baselines at test time, as ultimately this is what matters to the user. Our method is at least 2 orders of magnitude faster than the baseline in neural function evaluations (NFEs). However, as we detail next, our method is also orders of magnitude faster than the baselines in training time.
>
> * **Training time**: The diffusion-based baselines we compared against are all zero-shot methods that rely on a pre-trained unconditional diffusion prior (DDPM). Hence, the “effective” training time of these methods is the time it took to learn the DDPM prior, which for CelebA-HQ $256\\times 256$, Ho et al. stated it was $\\approx$ 63 hours on **8** V100 GPUs. For comparison, training our method (from scratch) on a CelebA-HQ $256\\times 256$ restoration task took $\\approx$5 hours on a **single** A6000 GPU (see Appendix A.3).
>
> * **Testing time**: As reported in Table 1, our method is at least 2 orders of magnitude faster than the baselines.  As mentioned in checklist item 8 (Experiments Compute Resources), we reported neural function evaluations (NFEs) as an architecture-agnostic measure. However, our architecture is significantly lighter than the U-net used in the compared posterior sampling baselines (both in terms of compute and memory footprint). Therefore, the numbers in Table 1 are actually underestimating the speedup factor introduced by our method. Putting aside the lower memory footprint, a single forward pass with a batch of 1 on an A6000 GPU with our architecture lasts 5$\\pm$0.1 ms, compared to 20$\\pm$0.4 ms for the U-Net used by DDRM/DDNM/RePaint, and 140$\\pm$8 ms for MAT. To ensure this is clear enough, in the camera-ready version we will include another column in Table 1 translating NFEs to runtime in seconds further emphasizing the speed advantage of posterior trees. (The rebuttal PDF includes Table 1 with runtime reported in GPU seconds).
>
> **Required hyper-parameter tuning to stabilize training**
> As mentioned in Section 3.4 L179, we observed that fixing the hyperparameters $\\varepsilon\_0=1,t\_0=5$ worked well for all experiments, and did not need to optimize these further. In general, it might be that for some tasks this strategy is suboptimal, requiring more careful tuning. Nonetheless, given that our training time is relatively short, this added computational burden is manageable, and it is still beneficial to use our method due to its much faster inference at test time.
>
> **Scaling to wider/deeper trees**
> Indeed, as mentioned in Section 5 (Discussion and Conclusion), our method is limited by the number of output leaves as we amortize the inference of the entire tree to a single forward pass. However, note that this limitation also exists for the baselines. To compute the baseline trees, we need to perform a two-step procedure: (1) Sampling $N\_s=100$ images from the posterior, and (2) performing hierarchical $K$-means $d$ times to build a tree of degree $K$ and depth $d$. In our experiments with $K=3,d=2$ (i.e. a tree with 9 leaves), we noticed that using less than $N\_s=100$ images often led to degenerate trees with one or more leaves having 1 sample or less. For example, consider the case of using $N\_s=9$ samples. Even if the posterior is perfectly balanced (often not the case), each leaf will have only 1 sample to “average” at depth 2\. Therefore, if we want to use the baselines for trees with significantly more leaves, we need to sample enough images ($N\_s\\gt 100$) per test input which scales poorly as well.
>
> Ultimately, the goal is to present the user with only a **few** representative prototypes **summarizing** the different options. Otherwise, navigating the underlying possibilities becomes tedious and time-consuming as the user has to skim through many images for every input.

---

> > ### Comment · Reviewer_VHHJ · 2024-08-12
> >
> > I appreciate the thorough response from the authors. My concerns have been addressed. I will increase my score.

---

### Official Review · Reviewer_7Dwv · 2024-07-12

**Soundness:** 4
**Presentation:** 4
**Contribution:** 4
**Rating:** 7
**Confidence:** 3

**Summary:**

This work proposes a technique to predict a tree-structured hierarchical summarization of a posterior distribution using a single forward pass of a neural network. The technique is an amortized hierarchical version of the oracle loss in multiple choice learning. Experiments show the method is effective at hierarchically representing the posterior both qualitatively and quantitatively.

**Strengths:**

1. The method is simple, efficient, and sound.
2. It addresses a fundamental and practically relevant problem (hierarchically visualizing complex distributions).
3. The presentation of the method and the experiments is excellent.

**Weaknesses:**

1. Table 1 reports number of function evaluations, but it's not clear how that translate to runtime since the time per function call and extent of parallelization varies .
2. The qualitative evaluation in Figure 1 and Figure 4 are not very informative. It's often hard to tell the differences between many of the nodes or judge whether there is an unambiguous underlying hierarchical structure.
3. The clustering / hierarchy produced the method is based on $L_2$ distance, which is not a very useful metric in many applications including images. Though as the authors discussed, this can be potentially addressed e.g. by first embedding the inputs with an autoencoder. Empirically demonstrating that this limitation can be addressed would be important for adoption of this method in many applications.

**Questions:**

1. Can you apply this method when the reconstruction loss is not an $L_2$ loss but a generic function?
2. How does the runtimes compared in Table 1?
3. Can you show how the methods compare as a function of runtime in Table 1? It's possible that you don't need to run that many function evals for the baselines to get similar performance.

**Limitations:**

The authors discussed limitations of the paper.

---

> ### Author Rebuttal · Authors · 2024-08-06
>
> **Comparing runtime in neural function evaluations vs GPU seconds**
> Please see Table 1 in the rebuttal PDF where runtime is reported in seconds. As mentioned in checklist item 8 (Experiments Compute Resources), we reported neural function evaluations (NFEs) as an architecture-agnostic measure. However, our architecture is significantly lighter than the U-net used in the compared posterior sampling baselines (both in terms of compute and memory footprint). Therefore, the numbers in Table 1 are actually underestimating the speedup factor introduced by our method. Putting aside the lower memory footprint, a single forward pass with a batch of 1 on an A6000 GPU with our architecture lasts 5$\\pm$0.1 ms, compared to 20$\\pm$0.4 ms for the U-Net used by DDRM/DDNM/RePaint, and 140$\\pm$8 ms for MAT. To ensure this is clear enough, in the camera-ready version we will include another column in Table 1 translating NFEs to runtime in seconds further emphasizing the speed advantage of posterior trees.
>
> **Baselines performance as a function of compute**
> Note that to compute the baseline trees we need to perform a two-step procedure: (1) Sampling $N\_s=100$ images from the posterior, and (2) performing hierarchical $K$-means $d$ times to build a tree of degree $K$ and depth $d$. Therefore, saving computation in this process requires sampling fewer images $N\_s$ per test input. However, in our experiments with $K=3,d=2$ (i.e. a tree with 9 leaves), we noticed that using less than $N\_s=100$ images often led to degenerate trees with one or more leaves having 1 sample or less. For example, consider the case of using $N\_s=9$ samples. Even if the posterior is perfectly balanced (often not the case), each leaf will have only 1 sample to “average” at depth 2\. In fact, it is likely that to achieve the optimal performance with the baselines more than $N\_s=100$ samples are needed, which would be even more computationally demanding. To better clarify this point, we will include an additional appendix discussing the success probability of building a tree with $K^d=9$ leaves as a function of the number of sampled images $N\_s$.
>
> **Learning posterior trees with other loss functions**
> This is a great point. A distinction should be made between the measure used for the clustering (i.e. determining the associations of samples to clusters) and the loss used within each cluster to determine the cluster representative. Our approach can work without modification with any association measure (e.g. LPIPS, some domain-specific classifier, etc.). However, changing the within-cluster loss requires some modifications. Specifically, the use of the $L\_2$ loss is what provides us with the hierarchical decomposition of the posterior mean (see Eqs. (2)-(7)). In particular, when using the $L\_2$ loss, each cluster representative becomes the posterior cluster mean, and a weighted combination of those representatives gives the overall posterior mean (which is the tree root). This allows us to have the network output only the leaves of the tree, which implicitly define the entire tree (as the nodes of each level are obtained as linear combinations of the nodes of its children).
>
> We will add an appendix discussing the distinction between association losses and within-cluster losses, and will include an illustration of using an association loss that is not $L\_2$. As for generalizing the method to work with other within-cluster losses, this is a great avenue, which we leave for future research.

---

> > ### Comment · Reviewer_7Dwv · 2024-08-08
> >
> > Thank you for the clarifications. I don't see the architecture-agnostic nature of NFE necessarily as a strength at all since what matters in practice is the runtime (and memory considerations), which depends not only on architecture, but also hardware utilization of each method. Regarding the provided runtime table, why did you choose to use a batch size of 1? Using a larger batch size should lead to significant speed up for the baselines since sampling is parallelized.

---

> > > ### Author Response · Authors · 2024-08-09
> > > **Runtime detailed comparison**
> > >
> > > Indeed, memory footprint and hardware parallelization ultimately determine the overall runtime. We tried to refrain from factoring these in our calculation as these numbers are hardware-specific depending on the available GPU. Nonetheless, our method also has a lower memory footprint and is just as amenable to parallelization. We apologize if our previous answer needed to be clearer. The table below benchmarks the speed of the forward pass and the GPU memory usage as a function of the batch size on an A6000 GPU with 48 GB:
> > >
> > > |                  |       | Ours  |       |&#124;       | DDRM/DDNM/RePaint |       |&#124;        |  MAT  |       |
> > > | :---:            | :---: | :---: |  :---: |  :---: |       :---:       | :---: |  :---: | :---: | :---: |
> > > | Batch size       |   1   |  10   |  430  |   1   |         10        |   97  |   1   |   10  |   30  |
> > > | forward pass (ms)|14$\pm$0.2|39$\pm$0.2|1150$\pm$2|34$\pm$0.4|245$\pm$0.4|2315$\pm$6.5|150$\pm$8|867$\pm$3.8|2599$\pm$13.6|
> > > | GPU memory (GB)  |  1.3  |  2.1  | 46.9  |  1.85 |        7.5        |  46.8 | 3.0   | 16.0  | 47.2  |
> > >
> > > For each method, we tested 3 different batch sizes: 1, 10, and the maximal number of samples that fit in memory. The forward pass speed and the memory usage in each setting were tested 100 times using CUDA events (reported as mean$\\pm$std). Our method is extremely fast and parallelizable, enabling the inference of 430 test images with a single forward pass of $\approx$1.15 seconds.
> > >
> > > **How does this compare to the baselines for a single test image?**
> > > Our method is far superior even for a single test image. For simplicity let us assume the cost of running hierarchical $K$-means is negligible, and that we can squeeze a batch size of $\approx$100 samples for the diffusion-based samplers, and $\approx$“33.33” samples for MAT. Sampling $N\_s=100$ posterior samples with DDRM (fastest diffusion baseline, requires only 20 denoising steps) lasts 46.3 seconds, and with MAT lasts 7.8 seconds. In comparison, our method requires only 0.014 seconds, leading to a $557\\times$ speedup compared to the fastest baseline.

---

### Official Review · Reviewer_4WsD · 2024-07-18

**Soundness:** 3
**Presentation:** 3
**Contribution:** 3
**Rating:** 7
**Confidence:** 3

**Summary:**

This work proposes to solve the problem of quantifying and visualising the uncertainty in the solutions of ill-posed inverse problems like image-to-image translation, image re-construction, inpainting, etc. The paper proposes to do so by using 'Posterior-trees' where the authors make use of the result that optimizers of Eq. (1) form CVT of the posterior. Further application of simple Bayes rule and simple probability allows for construction of a hierarchal tree to quantify as well as visualize the uncertainty associated with the predicted solution of the inverse problem at hand.

**Strengths:**

1. The paper is well written with clear ideas and rationale.
2. The idea of using hierarchal trees to quantify and visualize the uncertainty associated with the posterior of inverse problem is indeed novel. Although the core idea on which the paper builds upon is already presented in [45].
3. Nonetheless, extending the results of [45] to construct a heirarchal version of it is notable.
4. The results and visualizations are satisfying.

**Weaknesses:**

1. I think the authors should provide more elaborate background on CVT and results of [45], either in main text or in supplementary for ease of the reader.
2. One thing that I did not understand is - what is the trade-off between breadth and depth of the constructed tree? For example - what is the difference between having more children nodes with low depth and having less children nodes with high depth. This would help in further understanding the advantages and limitations of the proposed method.
3. Another point that I am a bit skeptical about is that the method operates directly in the image space (or the space of the input of the problem at hand). In case of images, this space is very high-dimensional. In such high-dimensional spaces, capturing all the variations of the solution specially through a discrete tree-like structure is problematic. I don't know how the proposed method is able to handle it.

Overall I enjoyed reading the paper, and the paper certainly seems to be novel, albeit, incrementally. Hence, I lean for weak acceptance.

**Questions:**

See weaknesses

**Limitations:**

See weaknesses

---

> ### Author Rebuttal · Authors · 2024-08-06
>
> **Providing more background on CVT and the results of \[45\]**
>
> We thank the reviewer for bringing this to our attention. In the camera-ready version, we will include an additional appendix summarizing the main results of \[45\] upon which we build.
>
> **Tradeoff between breadth and depth of the constructed trees**
>
> Kindly note that we discussed and visualized this tradeoff for the task of digit inpainting in Appendix F and Figs. A7-A8. We apologize for not referring to this appendix from the main text. We’ll add a reference, thanks for catching this.
>
> In short, the choice of $K$ and $d$ determines the layout of the output space tessellation. The degree $K$ controls the emphasis/over-representation devoted to weaker posterior modes. A smaller degree leads to more emphasis on weaker modes of the posterior as tree depth $d$ grows. However, the optimal layout $K$ and $d$ is task and input-dependant, and setting these adaptively is an interesting direction for future research.
>
> **Number of prototypes and operating in input/pixel space**
>
> This is a very good point. Please note that ultimately the goal is to present the user with only a **few** representative prototypes **summarizing** the different options. Otherwise, navigating the underlying possibilities becomes tedious and time-consuming as for every input the user has to skim through many many images. In our case, we chose these prototypes to be the cluster centers in pixel space at different levels of granularity. However, as we mentioned in Section 5 (Discussion and Conclusion), averaging in pixel space might lead to off-manifold “blurry” reconstructions, which could (possibly) be tackled by applying posterior trees in the latent space of an autoencoder such as VQ-VAE \- an intriguing direction for future work.
>
> **Novelty**
>
> While our approach is conceptually indeed a hierarchical extension of \[45\], as the reviewer noted, our work has several non-trivial contributions compared to \[45\], some of which are unique to the setting of working with tree structures:
>
> * Proposing an efficient architecture that enables predicting **diverse** solutions (and their likelihoods) compared to the fully shared (likelihood-free) architecture of \[45\], as we verify in Appendix A (Figs. A1-A2).
> * Preventing tree collapse and stabilizing the optimization steps through a novel adaptive regularization scheme and a weighted sampler that ensures proper leaf optimization with Adam (Appendices B-C, Figs. A3-A4).
> * Demonstrating, to the best of our knowledge, the first successful application of MCL with diverse results for high-resolution image-to-image regression tasks.

---

> > ### Comment · Reviewer_4WsD · 2024-08-13
> > **Response to Rebuttal**
> >
> > Thank you for your response. I am convinced with the response to my questions, further the response provided by the authors to Reviewer VHHJ helped me in further clarification. I am increasing my score accordingly!

---

### Author Rebuttal · Authors · 2024-08-07

Updated runtime table in seconds

---

### Decision · Program_Chairs · 2024-09-25

**Decision:**

Accept (poster)

**Comment:**

The paper presents a novel method for quantifying and visualizing uncertainty in solutions to ill-posed inverse problems using a tree-valued hierarchical summarization of the posterior distribution, achieved in a single forward pass of a neural network, allowing for efficient and granular exploration of solution spaces. Reviewers appreciated the paper's excellent presentation, the novelty and practical relevance of the hierarchical approach to visualizing complex distributions, and the method is qualitatively and quantitatively effective. Reviewers noted the need for more background on foundational concepts, concerns about the method's handling of high-dimensional data, potential limitations in scaling the tree depth and width due to architectural constraints, and the dependency on several heuristics for optimal performance. The reviewers responded positively to the authors' replies and unanimously recommended that the paper be accepted. Therefore, we accept the paper.